# MEMFREEZING: TOWARDS PRACTICAL ADVERSARIAL ATTACKS ON TEMPORAL GRAPH NEURAL NETWORKS

## ABSTRACT

Temporal graph neural networks (TGNN) have achieved significant momentum in many real-world dynamic graph tasks, making it urgent to study their robustness against adversarial attacks in real-world scenarios. Existing TGNN adversarial attacks assume that attackers have complete knowledge of the input graphs. However, this is unrealistic in real-world scenarios, where attackers can, at best, access information about existing nodes and edges but not future ones at the time of the attack. Nevertheless, applying effective attacks with only up-to-attack knowledge is particularly challenging due to the dynamic nature of TGNN input graphs. On the one hand, graph changes after the attacks may diminish the impact of attacks on the affected nodes. On the other hand, targeting nodes that are unseen at the attack time introduces significant challenges. To address these challenges, we introduce a novel adversarial attack framework, MemFreezing, to yield long-lasting and spreading adversarial attacks on TGNNs without the necessity to know knowledge about the post-attack changes in the dynamic graphs. MemFreezing strategically introduces fake nodes or edges to induce nodes' memories into similar and stable states, which we call the 'frozen state.' In this state, nodes show limited responses to graph changes and are compromised in their ability to convey meaningful information, thereby disrupting predictions. In subsequent updates, these affected nodes maintain and propagate their frozen state with support from their neighboring nodes. The experimental results demonstrate that MemFreezing can persistently decrease the TGNN models' performances in various tasks, delivering more effective attacks under practical setups.

## 1 INTRODUCTION

Dynamic graphs are prevalent in real-world scenarios, spanning areas like social media (Kumar et al., 2018), knowledge graphs (Leblay & Chekol, 2018), autonomous systems (Leskovec et al., 2005), and traffic graphs (Pareja et al., 2020). Unlike static graphs, whose nodes and edges remain constant, dynamic graphs evolve over time, introducing challenging tasks like link prediction and node classification on dynamically changing nodes and edges. Driven by successes of Graph Neural Networks (GNNs) (Kipf & Welling, 2016; Hamilton et al., 2017; Veličković et al., 2017; Xu et al., 2018), Temporal Graph Neural Networks (TGNNs) have emerged as state-of-the-art solutions in many dynamic graph tasks (Trivedi et al., 2019; Kumar et al., 2019; Rossi et al., 2020; Zhang et al., 2023; You et al., 2022). As such, there is a pressing need to study their robustness towards adversarial attacks, especially since such attacks have shown significant efficacy against traditional GNNs (Wang et al., 2018; Tao et al., 2021; Zügner et al., 2018; Zou et al., 2021; Ma et al., 2020; Zang et al., 2020; Bojchevski & Günnemann, 2019; Sun et al., 2022; Li et al., 2022). By modifying the input graphs with imperceptible and subtle perturbations, the adversarial attacks can inject noises into GNNs, making the models yield incorrect or adversary-expected results. For instance, a social media such as REDDIT Kumar et al. (2018) may employ TGNN to decide whether comments (as edges) from users to posts (as nodes) should be banned based on his/her comment histories. With subtle adversarial attacks, malicious messages can easily bypass this checking functionality.

While several studies have explored the effectiveness of adversarial attacks on dynamic graphs (Lee et al., 2024; Sharma et al., 2022; 2023; Chen et al., 2021), they often assume that attackers have complete knowledge of the input graphs at the time of the attack, which is impractical in many real-world scenarios. Specifically, as dynamic graphs evolve, by the time attackers observe the entire

evolution (i.e., track all changing nodes and edges) and identify the optimal timestamps to inject adversarial perturbations (e.g., adding fake nodes or edges), those key timestamps may have already passed, making it impractical to inject noises timely. As the old saying goes, 'It is easy to be wise after the event.' Therefore, in real-world cases, the adversarial may attack TGNN with only limited knowledge up to the time of the attack. Nevertheless, studying TGNN adversarial attacks under practical constraints is valuable, as real-world attacks with limited knowledge may exhibit unique characteristics that uncover vulnerabilities in TGNNs, which are overlooked when focusing solely on idealized scenarios.

However, attacking TGNNs with limited knowledge up to the attack time faces significant challenges due to the evolving nature of dynamic graphs. First, the impact of adversarial noise can quickly decay as the graph evolves and node information updates. Second, it is difficult to influence unseen nodes or edges that appear after the attack, as their information is unknown. Thus, an effective strategy must endure the graph's evolution and affect both current and future nodes despite this uncertainty.

Interestingly, the node updating mechanism in Temporal Graph Neural Networks (TGNNs) offers unique potentials for persisting and propagating adversarial noises in dynamic graphs. Generally, TGNNs maintain and update node status vectors, often referred to as *node memory* by recent studies (Rossi et al., 2020; Zhou et al., 2022; Wang & Mendis, 2024; Zhou et al., 2023; Wang & Mendis, 2023), to capture temporal history, which is crucial for delivering accurate predictions in dynamic graph tasks. Moreover, a node's memory vector can potentially affect its neighbors. When graph changes occur—such as the addition or deletion of nodes or edges—the memory vectors of related nodes are updated based on their neighbors' memories. This raises intriguing questions: *Can TGNN predictions be disrupted by disabling their memories, and can this effect persist and spread through their memory updates?*

To address this inquiry, we thoroughly investigated the memory update patterns of nodes within TGNNs and made the following observations: (1) Although it is not possible to directly affect unseen predictions, we can degrade TGNN prediction accuracy by pushing nodes—whether seen or unseen—into a relatively 'frozen' state, their memories remain stable and exhibit limited responsiveness to surrounding changes, reducing their ability to convey updated or meaningful information. (2) While a noisy node's memory vector may struggle to maintain its noisy state over time on its own, this state can persist for much longer if its neighboring nodes have similar memories.

We introduce MemFreezing, a novel adversarial attack designed to study the vulnerabilities of TGNNs under realistic constraints. At a specific attack timestamp, MemFreezing strategically selects groups of victim nodes that mutually reinforce each other's noisy states during updates, leveraging a heuristic we term 'cross-freezing'. By injecting carefully crafted fake messages, MemFreezing guides these nodes into a stable *frozen state*, where their memory updates exhibit high similarity over time. This stability reduces their responsiveness to graph changes and limits their ability to convey meaningful information, thereby misleading predictions. Additionally, we simulate future nodes to induce the propagation of adversarial noise. We summarize our contributions as follows:

- We recognize the limitations of existing adversarial attacks on dynamic graph models and identify the challenges of persisting and propagating adversarial noise in real-world threat models, particularly in Temporal Graph Neural Networks.
- We propose MemFreezing, an adversarial attack that disables node memories in TGNNs by pushing them into unnaturally stable states. To achieve this, we design a cross-freezing mechanism that induces nodes to be stable despite future updates and encourages affected nodes to propagate stable states by their simulating future neighbors.
- We compare our method with prior GNN adversarial attacks on various dynamic graphs. Experimental results show that MemFreezing effectively and persistently misleads TGNN predictions across diverse datasets and models, outperforming state-of-the-art GNN attacks, even in the presence of defenses.

## 2   BACKGROUND AND RELATED WORK

**Dynamic Graphs.** Unlike a static graph, a dynamic graph consists of nodes and edges evolving over time. Dynamic graphs can be represented in two ways: Discrete-Time Dynamic Graphs (DTDGs) describe dynamic graphs as a series of static snapshots taken periodically, while Continuous-Time

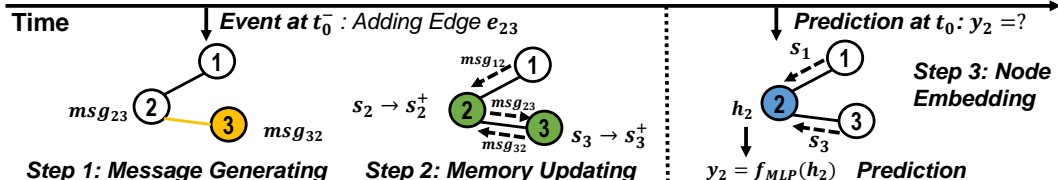

Figure 1: The three steps of TGNN computing assuming a new event at timestamp $t_0^-$ adds an edge $e_{23}$ to the dynamic graph: Firstly, messages $msg_{23}$ are generated for the nodes involved in this event nodes 2 and 3. Next, the nodes aggregate messages from their neighbors and update their memories (e.g., $s_2 \to s_2^+$). At a future prediction time $t_0$, nodes aggregate memories (e.g., $s_1$ and $s_2$) from their neighbors and embed them into node vectors (e.g., $h_2$) for the prediction.

Dynamic Graphs (CTDGs) view the graph as a collection of events—each event detailing updates like node or edge changes. Recent TGNNs focus on CTDGs since they can retain more information than DTDGs' fixed intervals and more complex (Kazemi et al., 2020). Within the CTDG paradigm, the dynamic graphs are represented as $G = \{x(t_1), x(t_2), ...\}$, in which $x(t_i)$ indicates an event happened at timestamp $t_i$. Generally, the prediction task for CTDGs can be depicted in equation 1.

$$y_i = f_\theta(G_i, t_i) = f_\theta(\{x(t_1), x(t_2), ...x(t_{i-1})\}, t_i) \tag{1}$$

At the prediction time $t_i$, the model $f_\theta(\cdot)$ takes all previous events $G_i = \{x(t_1), x(t_2), ...x(t_{i-1})\}$ as inputs and predicts the testing nodes' classes or future edges.

**Temporal Graph Neural Networks.** The memory-based Temporal Graph Neural Networks (TGNN) are widely studied and achieve state-of-the-art accuracies in dynamic graph tasks (Trivedi et al., 2019; Kumar et al., 2019; Rossi et al., 2020; Kazemi et al., 2020; Zhang et al., 2023; You et al., 2022). Generally, these TGNNs maintain node features across different timestamps that track the node's history and use it for predictions. Note that, despite their different names (e.g., node memories (Rossi et al., 2020; Wang et al., 2021), node representations (Trivedi et al., 2019), node dynamic embeddings (Kumar et al., 2019)) across various TGNNs, these node features are represented as vectors on presented nodes and evolve over time to capture the temporal information of these nodes. Following existing general TGNN frameworks (Zhou et al., 2022; 2023; Wang & Mendis, 2024; 2023; Rossi et al., 2020), we refer to these evolving node feature vectors as ***node memories***. As illustrated in Figure 1, TGNNs produce node embedding for the predictions in three steps. When an event $x(t_i)$ adds an edge $e_{uv}$ from node $u$ to node $v$ (i.e., $x(t_i) = e_{uv}$), two messages are generated as equation 2. For simplicity, we only present the updating and following operations of node $u$, which is the same for node $v$.

$$m_{vu} = msg(s_v, s_u, \Delta T, e_{uv}) \tag{2}$$

The $msg(\cdot)$ is a learnable function such as Multi-Layer-Perceptions (MLPs). The $s_u$ and $s_v$ denote the memories of node $u$ and node $v$ at their last updated times, and $\Delta T$ represents the difference between the current timestamp and the nodes' last updated times. Next, nodes $u$ and $v$ aggregate messages from their neighbors and update their memories as equation 3.

$$s_u^+ = UPDT(s_u, AGGR(m_{ku}|k \in N(u))), \tag{3}$$

The $N(u)$ denotes the neighbors of node $u$. The $AGGR(\cdot)$ is usually implemented by a $mean$ or $most\_recent$ function to aggregate messages from the node's neighbors (Rossi et al., 2020). The $UPDT(\cdot)$ uses the aggregated messages to update the node's memory and is usually implemented by a Gated-Recurrent-Unit (GRU) (Chung et al., 2014). When there is a prediction involving node $u$, TGNNs use a graph embedding module, such as Graph Attention Network (GAT) (Veličković et al., 2017), to embed the node's memory into the final node embedding, as depicted in equation 4.

$$h_u = GNN(s_u, s_k|k \in N(u)), \tag{4}$$

During prediction, TGNNs use nodes' latest memories (i.e., $s_i$ and $s_u$) to compute the node embedding $h_u$. The resulting node embedding $h_u$ is fed into an MLP for the final predictions.

**Adversarial Attacks on Graph Neural Networks.** The considerable achievements of GNNs have catalyzed numerous investigations into their resilience against adversarial attacks (Chen et al., 2017; Bai et al., 2018; Wang et al., 2018; Zügner et al., 2018; Bojchevski & Günnemann, 2019; Ma et al.,

Figure 2: The issues in TGNN adversarial attacks with unknown futures. At $t_0^-$, an attacker adds fake $node_1$ and $node_4$, which may effectively mislead predictions for $node_2$ and $node_3$ at $t_0$. However, as time progresses to $t_1^-$ and $t_2^-$, the appearance of $node_5$, $node_6$, and subsequent updates dilute the adversarial impact on $node_3$, making its prediction closer to the correct result (i.e., *Decaying* in the figure). Moreover, the attacks can hardly mislead predictions on new nodes coming after $t_0^-$ (i.e., $node_5$, $node_6$) since the attackers lack knowledge of these changes (i.e., *Missing* in the figure).

2020; Zang et al., 2020; Tao et al., 2021; Zou et al., 2021; Sun et al., 2022; Li et al., 2022; Zou et al., 2023). These adversarial attacks generally seek to misguide GNN predictions by modifying the nodes and edges of input graphs. For example, (Wang et al., 2018) introduces fake nodes with fake features that can minimize the loss between prediction results in the original graphs and the targeted fake results; (Zügner et al., 2020) adds and deletes edges that can cause the most substantial increases in the training losses on the original graphs. Recently, there have also been a few studies that explored the effectiveness of adversarial attacks on dynamic graphs and TGNNs Lee et al. (2024); Sharma et al. (2023; 2022); Chen et al. (2021).

## 3 PROBLEM ANALYSIS

### 3.1 THREAT MODEL

**Limits in Prior Threat Models.** Prior TGNN attacks (Lee et al., 2024; Sharma et al., 2023; 2022; Chen et al., 2021) assume that attackers have full knowledge of the target graphs and that these graphs remain static after the attacks. However, this assumption is impractical in real-world scenarios, as attackers cannot return to the optimal attack times after observing the entire evolution of a dynamic graph. In particular, when an attacker observes the evolution of a dynamic graph at $t_n$ and identifies optimal attack timestamps $t_{a_1}, t_{a_2}, ..., t_{a_k} \leq t_n$, they would need to go back to these past timestamps to inject noise, which is infeasible in practice. For example, if attackers aim to target a TGNN on social media like Reddit in October and determine that the optimal timestamps to inject noise were in September, they cannot go back a month to post adversarial comments or reviews that would influence node states in the underlying TGNN model.

**Threat Model.** Due to the limits discussed above, we assume that, for a practical and realistic TGNN adversarial attack, an attacker's knowledge is limited to events up to the attack timestamp, and the graph continues to evolve afterward. In particular, we set up our attack model as follows.

- *Attacker's Goal:* Given an evolving dynamic graph and a TGNN model, the attacker's goal is to misguide the TGNN predictions by introducing a limited amount of changes to the entire graph (e.g., affecting a small number of total nodes limited by the attack budget.)
- *Attacker's Knowledge:* Attackers can acquire knowledge only up to the attack's timestamp, including model details, presented inputs, and node memories, but not future changes in the dynamic graph. Regarding acquiring previous inputs, platforms like Wikipedia, Reddit, Meta, or X maintain dynamic graphs that adversaries can reconstruct with reasonable accuracy using publicly accessible data. Regarding acquiring the model details, many TGNN architectures and pre-trained models are open-sourced, and adversaries can also use techniques like insider threats or model extraction Yao et al. (2024); Oliynyk et al. (2023) to obtain model parameters if not publicly available. However, predicting future graph changes remains significantly harder. Therefore, we focus on the more challenging constraint of using only past knowledge for attacks while assuming a white-box setup for model parameters.
- *Attacker's Capability:* Attackers can add fake events as adding nodes/edges at the attack time. For example, while attacking TGNNs in social media, attackers can create fake user accounts as fake nodes and make junk comments to the blogs as fake edges. These fake events can impact the resultant node memories and embeddings and further influence predictions in TGNNs.

Figure 3: The resultant adversarial effects of the MemFreezing attack. (1) A node's frozen memory persists with support from its frozen neighbors, such as node 3 is kept frozen at $t_1^-$ and $t_2^-$. (2) The frozen states are propagated from affected nodes to their future neighbors, such as node 3 propagates to node 5 at $t_1^-$ and 6 at $t_2^-$. As a result, despite any post-attack changes, all associated predictions could be misguided since nodes' memories no longer work.

## 3.2 Motivation

**Issues of Existing Attacks.** Due to limited knowledge up to the attack time, adversarial attacks on TGNNs must contend with unknown future changes in dynamic graphs. However, the subsequent changes after the attack may significantly limit the attack performances for two reasons: Firstly, for seen and attacked targets, the noise that misleads their predictions becomes mixed with new information from future changes (as described in equation 3 and equation 4), making it too weak to mislead future predictions. Secondly, unseen nodes and edges added after the attack are difficult to affect, as the attackers have no knowledge of these future elements and cannot generate effective noise to mislead them. We illustrate examples of these issues in Figure 2. As details shown in Section 5 (e.g., Figure 6 and Table 1), while existing GNN attacks (Wang et al., 2018; Zou et al., 2021; Li et al., 2022) effectively reduce the model's accuracy immediately after the attack, they struggle to perturb predictions in the unknown future.

## 4 The MemFreezing Attack

We propose MemFreezing, an adversarial attack specifically tailored for TGNNs, to achieve our goals. It consists of two key features: i) To create long-lasting adversarial effects, we disturb TGNN predictions by freezing their node memories. Specifically, we induce nodes to mutually lock their memories, keeping them stable during future updates. As a result, the victim nodes become less responsive to surrounding changes, limiting their ability to provide critical information for predictions. ii) To affect unseen nodes and edges, we simulate future neighbors for the victim nodes and encourage these victim nodes to update the memories of their simulated future neighbors into similar, stable states. As a result, the adversarial effects remain persistent through future changes and influence subsequent predictions, as illustrated in Figure 3.

### 4.1 Freezing and Persisting Node Memory

**Memory Freezing Objective.** Instead of focusing on maximizing prediction losses as prior adversarial attacks, which are limited by unknown and diverse future events, we propose to transform victim nodes' memories into similar and stable states, which we refer to as *Frozen State*. In particular, by keeping node memories similar and unchanged over time, nodes in TGNNs can hardly carry or convey meaningful information, consequently disturbing predictions. To quantitatively investigate the potential effectiveness of freezing node memories, we freeze the node memories in TGN (Rossi et al., 2020) and JODIE (Kumar et al., 2019) by consistently forcing their node memories to all zero, then evaluate their performances on edge prediction tasks on Wikipedia (Kumar et al., 2018) dataset. As shown in Figure 4(a), this leads to significant accuracy drops over time, demonstrating the impact of freezing node memories. Thus, our attack objective is to freeze node memories into frozen states.

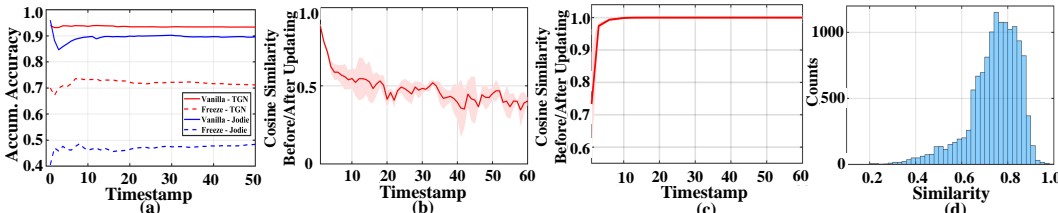

Figure 4: (a) The accumulated accuracy in vanilla TGNNs and their frozen counterpart. The ranges (colored bar) and averages (line) of the cosine similarities between node's evolving memories with (b) *persisting frozen by node themselves* and (c) *by connected groups (cross-freezing)*. (c) The distribution of cosine similarities among the ideal frozen states in different nodes.

**Challenges in Persisting Frozen Memories.** However, keeping nodes' memories in frozen states is challenging. Specifically, unpredictable messages aggregated from neighbors can considerably change a node's memory during updates, as described in equation 3. An intuitive solution is to maximize the contribution of a node's self-memory $s_u$ while minimizing the impact of incoming messages during its update. To explore the feasibility of this approach, we conducted a case study using TGN on the Wikipedia dataset. Specifically, we sample 100 victim nodes, manipulate each node's memory to block messages during updates, and then assess whether they remained unchanged over subsequent timestamps. As highlighted in Figure 4(b), despite this effort, the cosine similarities between nodes' memories before and after updating drop significantly over time. The primary challenge lies in the unpredictability of future messages, making it nearly impossible to find a memory that completely blocks new information from updates in the RNN or attention-based memory updating modules used by TGNNs. Further experimental details and theoretical proofs are provided in Appendix A.1.

**Opportunities in Freezing.** Although we can't block messages from a node's neighbors, we can introduce noise to those neighbors, making their messages help persist the noise. In particular, we assume that nodes' frozen memories in TGN can remain consistent if similar frozen neighbors surround them. We verify the assumption using the same model and data as Figure 4(b). Specifically, we first sample one-third of 100 victim nodes as *root node*, then sample two neighbors for each root node (referred to as *support neighbors*) and set their memories the same as the root node, then observe their memory changes over time. As depicted in Figure 4(c), if nodes have similar neighbors, their memories quickly converge to a relatively stable state and persist through future changes—we term this state as the node's *ideal frozen state*. Hence, we may expect sampled victim nodes to have similar frozen states so that they can mutually lock each other by then. Fortunately, as shown in Figure 4(d), the ideal frozen states from different nodes are similar; therefore, it is possible to keep nodes frozen by driving their memories into similar and stable states.

**Cross-Freezing Loss.** To this end, we propose to freeze victim nodes in connected groups and make them persist frozen with mutual support from each other. We termed this approach as ***Cross-Freezing***. Specifically, we would first sample a node as the root node, then sample two of its neighbors as support nodes—note that these support neighbors also cost our attack budgets—then force them to fulfill the following two goals: First, the nodes' memories should be similar so that the messages generated between them can potentially help to lock each other and keep their memory unchanged. Second, the nodes' memories should reach their ideal frozen states (i.e., the converged state in Figure 4(c)) so they cannot sense future changes after the attack. Hence, for each attacked node, we formulate the problem to be solved as equation 5.

$$\mathcal{L}_u^{freeze} = \sum_{k \in N_{\text{supp}}(u)} \left( \mathcal{L}_{\text{mse}}(s_k^*, s_k^+) + \mathcal{L}_{\text{mse}}(s_u^+, s_k^+) \right) \tag{5}$$

For any given node $u$ with its memory denoted as $s_u$ and support nodes as $N_{\text{supp}}(u)$, our objective relies on two Mean-Squared-Error (MSE) losses. The first, $\mathcal{L}_{\text{mse}}(s_k^*, s_k^+)$, aims to ensure that it updates its support neighbors' memory $s_k^+$ close to their ideal frozen state $s_k^*$. We find this state by updating the node's memory using current neighbors until they reach the stable state. Specifically, given a node and two of its support neighbors, we repeatedly update its memory using itself and its two support neighbors' memory until its memory is stabilized (i.e., has more than 0.9 cosine similarity before/after updates) or the maximum number of repeats is reached. The second loss,

represented by $\mathcal{L}_{\mathrm{mse}}(s_u^+, s_k^+)$, is designed to make sure that it updates its support neighbors' memory $s_k^+$ close to its own memory after updates (i.e., $s_u^+$).

## 4.2 Propagating Frozen States

**Future Simulating.** In addition to freeze nodes, it is also important to propagate the effect to more nodes to exacerbate attack effects, even though some nodes are unseen at the current stage. To make a node's memory influence unknown future neighbors, we propose using its existing neighbors to simulate potential future ones, which can then be used to optimize the victim nodes' ability to propagate their frozen state. This approach is based on the principle of homophily in real-world graphs, where neighboring nodes often exhibit strong similarities (McPherson et al., 2001). As an example, while applying TGN for the edge prediction tasks on Wikipedia dataset, nodes' neighbors have 0.87 cosine similarity on average, with over 60% neighbors having similarities over 0.9. Therefore, for a given node $u$, we use its current neighbors to simulate its future neighbors in two steps:

(1) First, we augment its current neighbor set $N(u)$ by adding 'similar fake future neighbors.' These are created by sampling the most recent ten neighbors from $N(u)$ (or fewer if there aren't ten) and introducing Gaussian noise. The noise has a mean of 0 for all nodes, and the standard deviation is set to 0.2 times the standard deviation of the current neighbor's memory. (2) Second, to simulate newly added nodes in the graph, we randomly add several 'newly presented fake future neighbors.' The memories of these fake neighbors are set to all-zero, similar to the initial memory for any new nodes. The number of these fake neighbors is determined by the ratio of newly appeared nodes among the most recent ten neighbors of the node. We use the $\Delta T$ of the most recent clean message on the victim nodes for these fake neighbors. For example, if we attack node n, whose most recent message before our attack appears at $\Delta T_k$, then the fake timestamp of adding those fake future neighbors will also be $\Delta T_k$. We also include more details about the future simulation in Appendix A.2 and further discuss its potential under extremely random and irregular graphs in Appendix C.15.

**Propagating Loss.** To make the frozen nodes contagious to potential future neighbors, we then use the resulting augmented neighbors $N_{\mathrm{aug}}(u)$ to solve the problem described in equation 6.

$$\mathcal{L}_u^{prop} = \sum_{k \in N_{\mathrm{aug}}(u)} \mathcal{L}_{\mathrm{mse}}(s_u, UPDT(s_k, m_{uk})) \tag{6}$$

The objective of this loss is to minimize the Mean Squared Errors (MSEs) between a node's memory and the memories of its new neighbors after an update. By doing so, we encourage the node's memory to update its current and potential future neighbors' memories to become similar to itself.

## 4.3 Attack Framework

Combining the above-mentioned solutions together, we introduce the two-stage MemFreezing attack framework as illustrated in Figure 5. We also included a detailed algorithm in Appendix B.

**Stage 1: Victim Node Selecting.** In this stage, we use a simple greedy approach to select the nodes to be attacked to facilitate cross-freezing. The connected victim nodes are select in two steps: First, we select the nodes with the highest degrees in the current graph as root nodes. The intuition behind this is that we want the injected noises to be propagated to as many nodes as possible, and these high-degree nodes, such as top-commented posts on social media, are usually popular in existing and future graphs. Next, for each root node, we select its two highest-degree neighbors as its support nodes. The following procedure will treat all the root and support nodes as victim nodes and transform them into frozen states.

**Stage 2: Adversarial Message Solving.** In this stage, we solve the messages to be passed to the selected victim nodes, which can be injected as fake nodes or edges. In the first step, we find the nodes' ideal frozen states (i.e., $s_u^*$ in equation 5) by updating its memory using current neighbors until convergence. In the second step, we simulate the future changes by augmenting victim nodes' neighbors with simulated futures (i.e., nodes/edges). The resulting neighbors are used as $N'(u)$ in equation 6. In the third step, we solve the adversarial memory $\hat{s}_u$ of the victim nodes by minimizing the total memory loss in equation 7. Specifically, for a node $u$, we solve its adversarial memory by minimizing the sum of its persisting (i.e., equation 5) and propagating (i.e., equation 6) losses.

$$\mathcal{L}_u = \mathcal{L}_u^{freeze} + \mathcal{L}_u^{prop} \tag{7}$$

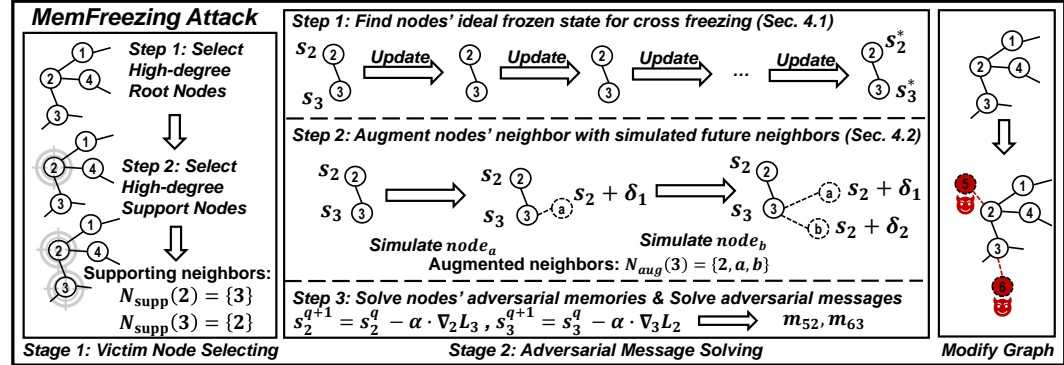

Figure 5: The two stages of the MemFreezing attack. In the **victim node selecting stage**, we greedily select victim nodes under the attack budget. In the **adversarial message solving stage**, we solve the victim nodes' targeted memory and corresponding adversarial messages. The solved messages are added to the graphs and removed after the attack timestamp.

Then, we solve the adversarial messages described in equation 8 so that these messages can update the nodes' memories into their solved frozen states.

$$\arg\min_{m_{\mathcal{A}u}} \mathcal{L}_{\text{mse}}(UPDT(s_u, AGGR(m_{\mathcal{A}u}, \widetilde{m_u}), \hat{s}_u)) \tag{8}$$

The $\widetilde{m_u}$ represents the aggregated messages collected from $u$'s other neighbors. In short, for node $u$, the solution aims to find a fake message $m_{\mathcal{A}u}$ that minimizes the MSE loss between the expected noise memory $\hat{s}_u$ and the memory updated after inserting it to the graph. Lastly, we add the solved noisy message as a fake node or fake edge for each victim node and remove it after the attack.

## 5 EVALUATION

### 5.1 EXPERIMENTAL SETUP

**Models and Datasets:** We use on four TGNN models for evaluation: JODIE (Kumar et al., 2019), Dyrep (Trivedi et al., 2019), TGN (Rossi et al., 2020) and Roland (You et al., 2022). The experiments use four dynamic graph datasets: Wikipedia (`WIKI`), Reddit (`REDDIT`) (Kumar et al., 2019), Reddit-body (`REDDIT-BODY`) and Reddit-title (`REDDIT-TITLE`) (Kumar et al., 2018). Details about the models and datasets are included in Appendix C.1. We also include results on a million-node dataset, `Wiki-Talk-Temporal` (Leskovec et al., 2010) in Appendix C.4.4.

**Tasks & Metrics:** We evaluate the models on two tasks: node classification and edge prediction (Rossi et al., 2020). For a timestamp, we measure the accuracy or area under the ROC Curve (ROC-AUC) based on all presented predictions from the beginning, which we termed as *accumulated accuracy* and *accumulated ROC-AUC*. More details about the tasks and matrices are in Appendix C.1

**Attack Setup:** We compare our work with three state-of-the-art GNN attacks: FakeNode (`FN`) (Wang et al., 2018), TDGIA(`TDGIA`) (Zou et al., 2021) and Meta-Attack-Heuristic(`Meta-h`) (Li et al., 2022). The results from Table 1 evaluate all attacks with 5% attack budgets, where we inject noises to 5% nodes of the input graph. In Appendix C.4, we evaluate attacks with 1% attack budgets. For our attack, we use a $1/3$ budget for the root nodes and $2/3$ for support nodes. All methods attack at the beginning of the test set (i.e., attack at $t_0$). We also include more details about the baseline attacks in Appendix C.2 and the results of injecting attacks in multiple timestamps in Appendix C.6.

**Defense Setup:** We adopt three adversarial defenses: Adversarial Training(`Adv_train`), Regularization under empirical Lipschitz (`Lip_reg`), and `GNNGuard` from static GNNs. More details about the defense setup are in Appendix C.3.

### 5.2 EXPERIMENTAL RESULT

**Overall Performance**. We examine the accumulated accuracy at three timestamps: $t_0 = 0$, $t_{25} = 25$, and $t_{50} = 50$. The results of the edge prediction task are presented in Table 1. As observed, all prior

Table 1: Accumulated accuracy of edge prediction in the vanilla/attacked TGNNs over different timestamps; lower matrices indicate more effective attacks. Results on more datasets and node classification tasks are included in Appendix C.4

| Dataset | | WIKI | | | | REDDIT | | | | REDDIT-BODY | | | |
|---|---|---|---|---|---|---|---|---|---|---|---|---|---|
| Model | | TGN | JODIE | Dyrep | Roland | TGN | JODIE | Dyrep | Roland | TGN | JODIE | Dyrep | Roland |
| Vanilla | | 0.93 | 0.87 | 0.86 | 0.94 | 0.97 | 0.98 | 0.96 | 0.95 | 0.90 | 0.87 | 0.90 | 0.88 |
| $t_0$ | FN | 0.81 | 0.74 | 0.74 | 0.82 | 0.84 | 0.83 | 0.84 | 0.83 | 0.76 | 0.82 | 0.77 | 0.79 |
| | Meta-h | 0.90 | 0.83 | 0.81 | 0.85 | 0.93 | 0.95 | 0.90 | 0.92 | 0.86 | 0.83 | 0.88 | 0.85 |
| | TDGIA | **0.77** | **0.72** | **0.71** | **0.80** | 0.74 | 0.80 | 0.81 | 0.74 | 0.72 | 0.81 | 0.74 | 0.76 |
| | Ours | 0.89 | 0.78 | 0.83 | 0.87 | 0.75 | 0.84 | 0.94 | 0.82 | 0.84 | 0.85 | 0.81 | 0.78 |
| $t_{25}$ | FN | 0.92 | 0.87 | 0.85 | 0.94 | 0.97 | 0.97 | 0.96 | 0.93 | 0.90 | 0.86 | 0.89 | 0.88 |
| | Meta-h | 0.93 | 0.87 | 0.84 | 0.93 | 0.96 | 0.98 | 0.94 | 0.96 | 0.89 | 0.86 | 0.90 | 0.87 |
| | TDGIA | 0.93 | 0.81 | 0.84 | 0.92 | 0.94 | 0.95 | 0.95 | 0.90 | 0.89 | 0.85 | 0.89 | 0.88 |
| | Ours | **0.80** | **0.75** | **0.77** | **0.85** | **0.81** | **0.84** | **0.91** | **0.80** | **0.81** | **0.84** | **0.76** | **0.80** |
| $t_{50}$ | FN | 0.94 | 0.87 | 0.86 | 0.94 | 0.97 | 0.97 | 0.96 | 0.95 | 0.90 | 0.86 | 0.90 | 0.88 |
| | Meta-h | 0.93 | 0.87 | 0.85 | 0.93 | 0.97 | 0.98 | 0.94 | 0.95 | 0.90 | 0.86 | 0.90 | 0.88 |
| | TDGIA | 0.93 | 0.87 | 0.85 | 0.93 | 0.96 | 0.97 | 0.95 | 0.92 | 0.89 | 0.86 | 0.90 | 0.87 |
| | Ours | **0.75** | **0.76** | **0.75** | **0.84** | **0.80** | **0.84** | **0.91** | **0.80** | **0.77** | **0.82** | **0.76** | **0.77** |

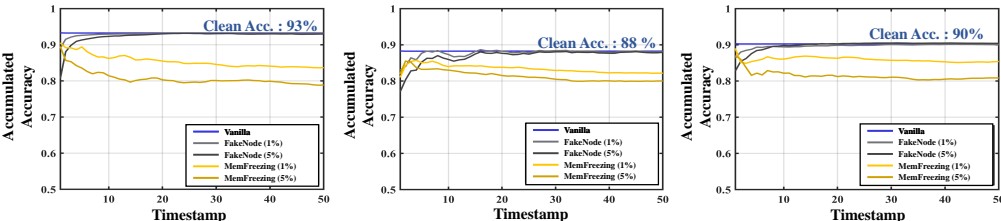

Figure 6: Accumulated accuracies of TGN under no defense(left), `Adv_train`(middle), and `Lip_reg`(right) with FakeNode and our attack on `WIKI` dataset. More results in Appendix C.5

attacks cause significant accuracy drops at $t_0$, but their impact quickly diminishes over time. By $t_{25}$ and $t_{50}$, the accumulated accuracy under these attacks is nearly identical to the baseline. In contrast, MemFreezing consistently disrupts model predictions. While it does not cause the largest accuracy drop at $t_0$ compared to other attacks due to the freezing objective, its effects are more persistent and even increase over time, achieving greater drops as the timestamps shift to $t_{25}$ and $t_{50}$. We also observe similar effects on MemFreezing when conducting the attack at different timestamps as detailed in Appendix C.7. The attacks on JODIE are less effective because JODIE employs a memory decay mechanism that uniformly decays previous memories. This introduces additional information outside the node memory, making JODIE more resilient to memory-based attacks. We discuss this phenomenon in more detail and explore potential defenses against MemFreezing in Appendix D.

**Performance under Defenses**. We further illustrate the accumulated accuracy of TGNs under different defenses in Figure 6. Similar results are observed: the effects of baseline attacks quickly diminish, resulting in only a 1.1% accumulated accuracy drop by $t_{50}$. In contrast, MemFreezing causes progressively larger accuracy drops over time, averaging over 10% drop by $t_{50}$.

**Ablation Studies.** To analyze the propagating and persisting capability of the noise solved by MemFreezing, we capture 100 victim nodes in TGN in edge prediction on `WIKI` and monitor the changes in their memory and their neighbors' memory. In Figure 7, we compare the cosine similarity between the memories of the victim nodes at $t_0$ with those in themselves and their one-hop and two-hop neighbors at each timestamp after the attack in four versions: (1) MemFreezing, (2) MemFreezing w/o frozen state (i.e., w/o using $\mathcal{L}_{mse}(s_k^*, s_k^+)$ in equation 5), (3) MemFreezing w/o cross-freezing loss (i.e., without using entire $\mathcal{L}_u^{freeze}$ in equation 5), and (4) original TGNN without attacks. The result shows that, in MemFreezing, the noise in the victim node can persist over ten timestamps, with over 0.92 cosine similarities. For the one-hop neighbors, at $t = 1$, they achieve 0.51 average similarities after the first update by the message from victim nodes, and at $t = 15$, the average rises to 0.88. The two hot neighbors, whose memories are updated by the message from one hot neighbor, have average similarities that grow from 0.24 to 0.84. In contrast, the similarity between nodes' initial attacked memory and their future counterparts drops drastically in the original TGNNs like (4). If the frozen states are not guaranteed like (2), the similarities also suffer drops and fail to achieve comparable similarities as (1). This is because, in such cases, the memories will change before reaching their converged states, making the final converged state different from the original adversarial memory states. Hence, this unchanged memory will harden the cross-freezing process,

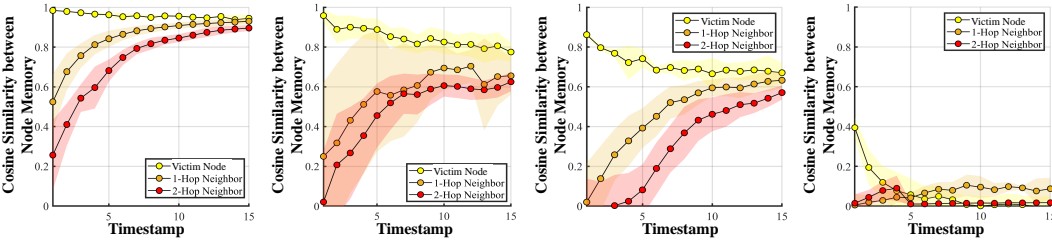

Figure 7: The similarities between victim nodes' initial noisy memories (at the time of the attack) and themselves'/their subsequent neighbors' memories in MemFreezing(left), MemFreezing w/o (middle-left) frozen state, MemFreezing w/o cross-freezing loss (middle-right), and regular nodes (right). All results above are from TGN and `WIKI`. More results are included in Appendix C.8.

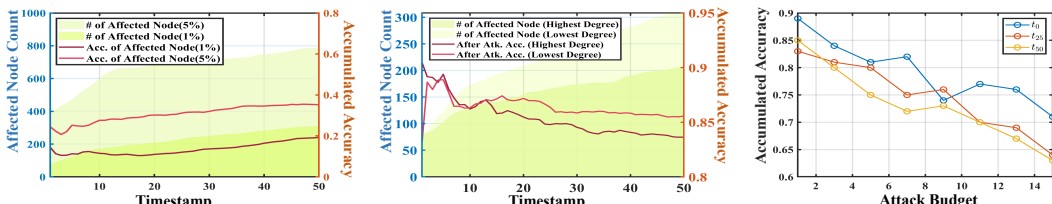

Figure 8: (left) Count of nodes affected by MemFreezing and accuracy for the affected nodes over time. (middle) Count of affected nodes and overall accuracy over time with two strategies for selecting the injected node: 1% lowest degree versus 1% highest degree nodes. (right) The accumulated accuracy at $t_0$, $t_{25}$, and $t_{50}$ under different attack budgets (% of total nodes). All results above are from TGN and `WIKI`. Results on more models and datasets are included in Appendix C.11.

which prefers similar memories among nodes. Therefore, the converged state is essential for persisting noisy memories. The similarities drop faster if we remove the cross-freezing loss like (3) since the cross-freezing mechanism is entirely disabled. Moreover, despite removing cross-freezing losses, the neighbors are getting more similar to the target nodes, indicating that the propagating loss works as expected. We also analyzed the advances of freezing node memories compared to maximizing prediction losses in Appendix C.9 and the stealthiness of the injected noises in Appendix C.10.

**Propagation in Dynamic Graphs.** To better understand how frozen effects spread in MemFreezing, we track all topologically connected nodes to the victim node, labeling them as affected nodes since noise can potentially propagate to them. We then measure the prediction accuracy of these affected nodes (represented by colored lines). As shown in Figure 8 (left), MemFreezing progressively impacts more nodes (green area) and significantly reduces prediction accuracy, even though some nodes were unseen at the attack timestamp. On the one hand, attacking high-degree nodes helps propagate the noises to more nodes, nearly doubling the number of affected nodes compared to low-degree targets. On the other hand, once nodes enter a stable (frozen) state, they propagate adversarial effects to future neighbors, ensuring the attack's persistence and adaptability despite dynamic graph changes. As shown, Figure 8 (middle) demonstrates that selecting high-degree victim nodes accelerates noise propagation and results in greater accuracy degradation compared to targeting low-degree nodes.

**Scale with Attack Budget.** We also evaluate MemFreezing under broader attack budgets ranging from 1% to 15%. As shown in Figure 8 (right), higher attack budgets lead to greater accuracy drops, demonstrating MemFreezing's scalability with increased attack costs.

## 6 CONCLUSION

In this work, we propose MemFreezing, a novel adversarial attack tailored for TGNNs, to overcome the challenges in attacking TGNN under limited-knowlegde scenarios. The MemFreezing attack misleads model predictions by freezing node memories in TGNNs into stable and dysfunctional states. The experimental results show that our approach can produce long-lasting and contagious noises in dynamic graphs, leading to significant performance drops in TGNNs.

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

APPENDIX

# A EXTENDED DESIGN

## A.1 SELF-FREEZING EXPERIMENTAL SETUP

We explored the viability of freezing a node by itself with a case study in TGN (Rossi et al., 2020), where the $UPDT(\cdot)$ function is typically realized using a GRU (Chung et al., 2014). At a particular timestamp, we randomly sample 100 nodes from the Wikipedia dataset and modify their memories. For each node, we use Adam optimizer (Kingma & Ba, 2014) to find a memory vector to suppress GRU updates by minimizing its reset gates (Chung et al., 2014). We then assessed if this memory state remains consistent over time.

The TGN used by the experiment uses GRU for memory updating (i.e., for implementing $UPDT(\cdot)$ function in equation 3), as depicted in equation 9-12.

$$r_t = \sigma(W_{ir}\widetilde{m_t} + b_{ir} + W_{hr}s_{t-1} + b_{hr}) \tag{9}$$

$$z_t = \sigma(W_{iz}\widetilde{m_t} + b_{iz} + W_{hz}s_{t-1} + b_{hz}) \tag{10}$$

$$n_t = tanh(W_{in}\widetilde{m_t} + b_{in} + r_t \odot \sigma(W_{in}\widetilde{m_t} + b_{in}) \tag{11}$$

$$s_t = (1 - z_t) \odot n_t + z_t \odot s_{t-1} \tag{12}$$

where $\sigma(\cdot)$ is the sigmoid function. Given the node memory $s_{t-1} \in \mathbb{R}^M$ at the previous timestamp, and the aggregated message $\widetilde{m_t} \in \mathbb{R}^D$ at time $t$, GRUs first compute reset gate $r_t \in \mathbb{R}^M$, update gate $z_t \in \mathbb{R}^M$, and new gate $n_t \in \mathbb{R}^M$.

In this experiment, we aim to minimize the interference of the message, $\widetilde{m_t}$, and maintain the updated memory, $s_t$, close to the previous memory, $s_{t-1}$. To this scope, we can maximize all the features in the update gate, $z_t$, until it approaches $\mathbb{1}$, where the update gate will be directly used to control the portion of the previous memory, which is:

$$\text{as } z_t \to \mathbb{1}, \quad s_t \to \mathbb{0} \odot n_t + \mathbb{1} \odot s_{t-1} \approx s_{t-1} \tag{13}$$

Additionally, according to Equation 10, the update gate $z_t$ is computed by the sum of two linear processes, and one is from the message, $\widetilde{m_t}$ and the other one is from memory $s_{t-1}$. As we maximize the linear output of the memory, $W_{hz} \cdot s_{t-1}$, the update gate, $z_t$, is then maximized.

Hence, to analyze the maximum output of the linear process, $W_{hz} \cdot s_{t-1}$, we formulate it into a linear program problem with the equations:

$$\begin{aligned} \max \; &\sum W_{hz} \cdot s_{t-1} \\ \text{s.t.} \; &-\mathbb{1} \le s_{t-1} \le \mathbb{1} \\ &W_{hz} \cdot s_{t-1} > \delta \end{aligned} \tag{14}$$

As the memory is the output of the $tanh$ function rather than the unit-length vector, $s_t$ is bounded by the limit of the $tanh$ function, $[-1,1]^M$. Further, we introduce an addition constraint $W_{hz} \cdot s_{t-1} > \delta$ to guarantee all dimensions of the linear output are bound by a constant, $\delta$.

The optimal result for the memory, $s_{t-1}^*$, for the linear problem only depends on the model weights, where given a TGN model, the solution of the self-freezing memory is unique, and we have conducted the experiment on three models TGN+WIKI, TGN+REDDIT, and a randomly initialized model.

The result in Figure 9-11 (a) shows the maximum update gate, $z_t^*$, computed by $\sigma(W_{hz} \cdot s_{t-1}^*)$. In the TGN+wiki example, $z_t^*$ is a 172-dimension vector, and it is distributed with a mean of 0.64 and a standard deviation of 0.12. As aforementioned, to achieve the self-freezing memory, the update gate, $z_t$, is required to approach $\mathbb{1}$, but it is infeasible to fine the solution in the real world case under the constraints. In Figure 9-11 (b), we simulate the GRU updating starting with the optimal memory, $s_{t-1}^*$, and monitor the cosine similarity between the memory before updated and after updated. The results further demonstrate even the optimal solution cannot accomplish the self-freezing goal.

To theoretically analyze the maximum of the in the general case, we divide them into their eigen-representations, and we use the SVD decomposition:

$$W_{hz} = U \cdot \Sigma \cdot V^T = \sum_i e_i \cdot U_i \cdot V_i^T, \quad s_{t-1} = \sum_{i|V_i \in V} \alpha_i \cdot V_i \tag{15}$$

In SVD decomposition, $U$ and $V$ are the unitary matrix, and we use the basis from $V$ to decompose $s_{t-1}$. Moreover, the linear process is written as:

$$W_{hz} \cdot s_{t-1} = \sum_i e_i \cdot \alpha_i \cdot U_i \cdot V_i^T \cdot V_i = \sum_i \alpha_i \cdot e_i \cdot U_i \quad (16)$$

This linear process is represented by the linear combination on the basis of $U$. We can easily acquire the theoretical maximum of the output. As $s_{t-1} \in [-1, 1]^M$, if $V_\theta$ is a basis of $\{-\frac{1}{\sqrt{M}}, \frac{1}{\sqrt{M}}\}^M$, $s_{t-1} = s_\theta$ can achieve the maximum projection to this basis, which is, $\alpha_\theta = \sqrt{M}$ and $s_{t-1} = V_\theta$. Similarly, the linear output $W_{hz} \cdot s_{t-1}$ achieves the maximum by there exist a basis $U_\theta = \{\frac{1}{\sqrt{M}}\}^M$, and the linear output,

$$W_\theta \cdot s_\theta = e_\theta \cdot \frac{1}{\sqrt{M}} \cdot \sqrt{M} \cdot \mathbb{1} = e_\theta \cdot \mathbb{1} \quad (17)$$

As is shown, the maximum output of the linear process is equal to the eigenvalue. According to the experiment, the largest eigenvalue of the weight matrix is usually around 2. Therefore, the update gate, $z_t$, has the theoretical maximum value, $\sigma(e_\theta) \approx 0.88$.

However, the weights, $W_{hz}$, are trained through the model update, which makes it impossible to find the ideal maximum in the practical case.

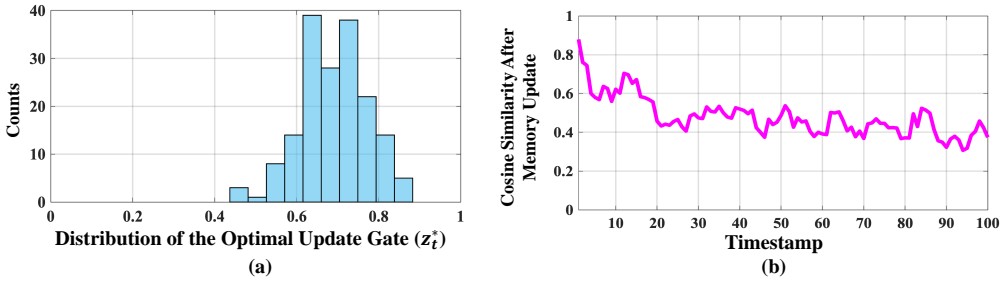

Figure 9: (a) The distribution of the optimal update gate $z_t^*$. (b) The cosine similarity between memory before the update and after the update, starting with the optimal self-freezing memory $s_t^*$. Experiments are conducted in the TGN model with `WIKI` datasets.

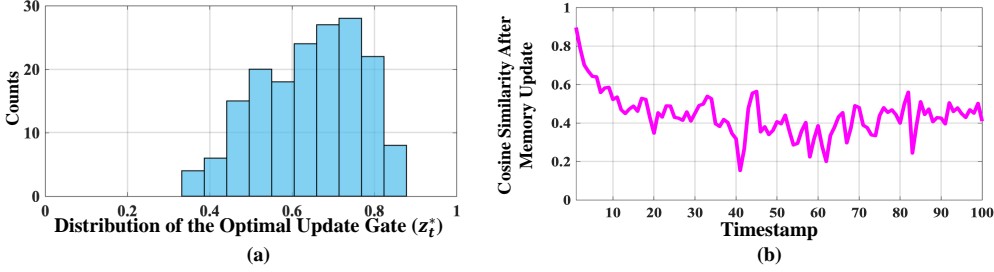

Figure 10: (a) The distribution of the optimal update gate $z_t^*$. (b) The cosine similarity between memory before the update and after the update, starting with the optimal self-freezing memory $s_t^*$. Experiments are conducted in the TGN model with `REDDIT` datasets.

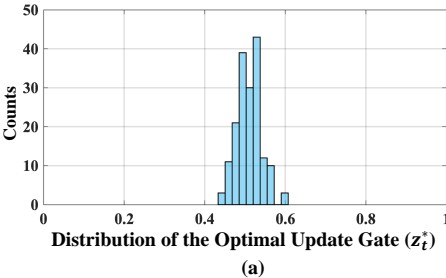 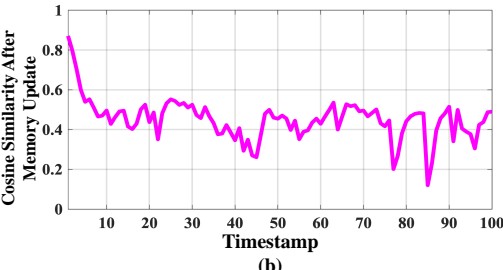

(a)                                         (b)

Figure 11: (a) The distribution of the optimal update gate $z_t^*$. (b) The cosine similarity between memory before the update and after the update, starting with the optimal self-freezing memory $s_t^*$. Experiments are conducted in the randomly sampled GRU model.

## A.2 DETAILS OF SIMULATING FAKE FUTURE NEIGHBORS

To simulate the potential future neighbors of the victim nodes and enhance their capability to contaminate those nodes, we randomly sample existing neighbors from victim nodes' and add Gaussian noise to their features. For the mean of the Gaussian noise, we use 0 as the mean for all nodes. We use 0.2 times the standard variation of the original neighbor's memory for the standard variations of the Gaussian noise. In summary, for a node $v$, we follow Equation equation 18 to simulate a victim node's neighbors,

$$s_i' = s_i + \mathcal{N}(0, \eta \cdot \sigma(s(v))), i \in N(v) \tag{18}$$

In which $s_i'$ stands for the fake future neighbors and $s_i$ stands for the memories from a sampled existing neighbor, $N(v)$ indicates the current neighbor set of node $v$. For the Gaussian noise $\mathcal{N}(0, \eta \cdot \sigma(s(v)))$, it has meant as 0, $\eta = 0.2$, and $\sigma(s(v))$ as the standard variation of all features in existing neighbors.

We use the $\Delta T$ of the most recent clean message on the victim nodes for the timestamp of their appearances. For example, if we attack node n, whose most recent message before our attack uses $\Delta T_k$ at its updating, then the timestamp of the fake future neighbors will also be $\Delta T_k$. It is also worth mentioning that, the $\Delta T$ has limited effects on the updating process. As proposed in TGAT and used in TGN and other TGNN models, the $\Delta T$ is encoded into a time vector first as

$$E(\Delta T) = W * cos(\Delta T)$$

in which $W$ is a weight vector with 172 dimensions with descending magnitudes (for example, $[1.00, 0.88, 0.78, ...1.12e - 09, 9.99e - 10]$). Then, $W$ is used to update the memory. The value of $W$ is very small except for the first few dimensions, making them can hardly affect the updating process.

## B  OVERALL ALGORITHM

---

**Algorithm 1: MemFrezzing Attack**

---

**Input**    : $\mathcal{G} = (V, s(V)) \leftarrow$ Original graph with Node $V$, memories $s(V)$
**Input**    : $\forall_{i,j|V_i,V_j \in V} \quad m(s_i, s_j, e_{ij}, \Delta t) \leftarrow$ Messages before $t_0$
**Input**    : $\mathcal{B} \leftarrow$ Number of attacked nodes (attack budget)
**Input**    : $q \leftarrow$ Number of support neighbors for each root node.
**Input**    : $N(V), N_{\text{supp}}(V), N_{\text{aug}}(V) \leftarrow$ the full neighbors sets, supported neighbors, and augmented
               neighbors set
**Output** : $V_{\mathcal{A}}, e_{\mathcal{A}}$: Perturbed nodes and message.

/* **Stage 1. Victim Node Sampling** */
$n \leftarrow \mathcal{B}/(q+1)$
$V_1^{\text{root}}, V_2^{\text{root}}, \cdots, V_n^{\text{root}} \leftarrow topk(\text{degree}(V), n)$
**for** $i \in \{1, 2, \cdots, n\}$ **do**
   $\mathbf{V}^{\text{support}} \leftarrow V_i^{\text{root}} \cup \{V_1, V_2, \cdots V_q \in N(V_i^{root})\}$
   $\{\hat{s}_0, \hat{s}_1, \cdots \hat{s}_q\} \leftarrow \texttt{ComputeConvergeState}(\mathbf{V}^{\textit{support}})$
   $\mathbf{V}_{\mathcal{A}}, \mathbf{e}_{\mathcal{A}} \leftarrow \texttt{ComputeAdversarialMessage}(\{\hat{s}_0, \hat{s}_1, \cdots \hat{s}_q\}, \mathbf{V}^{\textit{support}})$

/* **Stage 2. Solving Frozen State** */
**Function** $\texttt{ComputeConvergeState}(\mathbf{V}^{\textit{support}})$
   /* **2.1. Solving the Ideal Frozen State** */
   **for** $i \mid V_i \in \mathbf{V}^{\textit{support}}$ **do**
      $s_i \leftarrow s(V_i)$
      $m \leftarrow m(s_i, s_i, e_{ij}, \Delta t)$ , where $j \mid V_j \in \mathbf{V}^{\text{support}}$
      **do**
         $s_i \leftarrow s_i^+$
         $m \leftarrow m(s_i, s_i, e_{ij}, \Delta t)$
         $s_i^+ \leftarrow UPDT(s_i, m)$
      **while** $||s_i^+ - s_i||_2^2 > \epsilon$;
   $s_1^*, s_2^*, \cdots s_q^* \leftarrow s_1^+, s_2^+, \cdots s_q^+$

   /* **2.2. Solving the Cross-Frozen State** */
   $s_1^{(0)}, s_1^{(0)}, \cdots s_q^{(0)} \leftarrow s_1^*, s_2^*, \cdots s_q^* + \mathcal{N}(0, \eta \cdot \sigma(s(V)))$
   **for** $t \in \{0, 1, 2, \cdots, T\}$ **do**
      $\forall_{i \in \{1,2,\cdots q\}}, \quad s_i^{(t)+} \leftarrow UPDT(s_i^{(t)}, \widetilde{m_i})$
      **for** $i \in \{0, 1, 2, \cdots, q\}$ **do**
         $\mathcal{L}_i^{freeze} \leftarrow \sum_{k \in N_{\text{supp}}(i)} \left(\mathcal{L}_{\text{mse}}(s_k^{(t)+}, s_k^*) + \mathcal{L}_{\text{mse}}(s_i^{(t)+}, s_k^{(t)+})\right)$
         $\mathcal{L}_i^{prop} \leftarrow \sum_{k \in N_{\text{aug}}(i)} \mathcal{L}_{\text{mse}}(s_i^{(t)}, UPDT(s_i^{(t)}, m_{ik}))$
      $\forall_{i \in \{1,2,\cdots q\}}, \quad s_i^{(t+1)} \leftarrow s_i^{(t)+} - \alpha \cdot \nabla_{s_i}(\mathcal{L}_i^{freeze} + \mathcal{L}_i^{prop})$
   **return** $\{s_0^{(T)}, s_1^{(T)}, s_2^{(T)}, \cdots s_q^{(T)}\}$

/* **Stage 3. Solving the Adversarial Message** */
**Function** $\texttt{ComputeAdversarialMessage}(\{\hat{s}_0, \hat{s}_1, \cdots \hat{s}_q\}, \mathbf{V}^{\textit{support}})$
   **for** $V_i \in \mathbf{V}^{\textit{support}}$ **do**
      $V_{i,\mathcal{A}} \leftarrow V \mid V \in N(V_i)$
      **for** $t \in \{0, 1, \cdots, T\}$ **do**
         $m_{\mathcal{A}i}^{(t)} \leftarrow m(s_i, s(V_{\mathcal{A}}), e_{\mathcal{A}i}^{(t)}, \Delta t)$
         $\mathcal{L}_{\mathcal{A}} \leftarrow \mathcal{L}_{\text{mse}}(UPDT(s_i, AGGR(m_{\mathcal{A}i}^{(t)}, \widetilde{m_i})), \hat{s}_i)$
         $e_{\mathcal{A}i}^{(t+1)} \leftarrow e_{\mathcal{A}i}^{(t)} - \alpha \cdot \nabla_{e_{\mathcal{A}i}^{(t)}} \mathcal{L}_{\mathcal{A}}$
   **return** $\{V_{1,\mathcal{A}}, V_{2,\mathcal{A}}, \cdots, V_{q,\mathcal{A}}\}, \{e_{\mathcal{A}1}, e_{\mathcal{A}2}, \cdots, e_{\mathcal{A}q}\}$

---

# C    EXTENDED EVALUATION

## C.1    EXPERIMENTAL DETAILS.

**Model Details**. All four TGNNs we included maintain a memory vector in each node and follow the memory updating process as discussed in Section 2. And they are different in their node embedding procedure (i.e., equation 4). Specifically, Dyrep directly uses the node memories for the predictions (i.e., $h_i^t = s_i^t$). JODIE applies a time-decay coefficient to the scale memories before classification (i.e., $h_i^t = \delta(t) \cdot s_i^t$). TGN, on the other hand, refines memories using a single-layer graph attention module, as outlined in equation 4. Unlike prior models, ROLAND You et al. (2022) is a recent model designed for DTDG graphs, yet it also maintains a history node feature for each node as memory. Specifically, it adopts a multi-layer memory mechanism by keeping memory for both memory and embedding stages. In other words, for the graph embedding part, it also adopts a GRU to combine nodes' previous embedding with the current embedding gathered from updated node memories. All the models update and embed memory for one time at each prediction (i.e., one layer aggregation in equation 3 and equation 4). The node memory dimension is set to 172, and the node embedding dimension is set to 100. Following the training steps in (Rossi et al., 2020), we use Adam optimizer with learning rate $\alpha = 0.01$ to train the models 120 epochs.

**Tasks Details.** Models for node classification are trained to predict binary labels on each node. We use the commonly used Area under the ROC Curve (ROC-AUC) to measure the model performances. The models for edge prediction are self-supervise trained, using the edge information in future steps. During the testing, given a source node, they predict the possibility of whether another node will be its next incoming destination node and then decide which node will be its next neighbor. We use prediction accuracy for evaluating the edge prediction result.

**Dataset Details.** Reddit and Wikipedia are dynamic interaction graphs retrieved from online resources in (Rossi et al., 2020). In Wikipedia datasets, the nodes represent users and wiki pages, and the edges indicate editing from users to pages. In the Reddit dataset, the nodes represent users and subreddits, and an edge within it represents a poster from a user posted on a subreddit. The edge features are represented by text features, and the node labels indicate whether a user is banned. All the abovementioned information is accompanied by timestamps. Align with their original designs (Kumar et al., 2019), and we set the newly input nodes' features as zero feature vectors. Reddit-body and Reddit-title are two larger-scale datasets that represent the directed connections between two subreddits (a subreddit is a community on Reddit). The dataset is collected by SNAP using publicly available Reddit data of 2.5 years from Jan 2014 to April 2017 (Kumar et al., 2018). The statistics of the dataset used are shown in Table 2.

Table 2: Dataset details

|  | # of Nodes | # of Edges | # Edge Feature | # of Node Feature |
|---|---|---|---|---|
| **Wikipedia(`WIKI`)** | 9,227 | 157,474 | 172 | 172 |
| **Reddit(`REDDIT`)** | 11,000 | 672,447 | 172 | 172 |
| **Reddit-Body(`REDDIT-BODY`)** | 35,776 | 286,561 | 64 | 172 |
| **Reddit-Title(`REDDIT-TITLE`)** | 54,075 | 571,927 | 64 | 172 |

**Platform details.** We list then environment details in Table 3.

Table 3: Experimental Environment Setting

| Environment | Details |
|---|---|
| **OS** | Windows 11 |
| **CPU** | Intel i9-13900K |
| **Memory** | 64GB DDR5 RAM |
| **GPU** | NVIDIA RTX 4090 |
| **Platform** | PyTorch 2.2.1 |
| **CUDA Version** | CUDA 12.1 |

## C.2 BASELINE ATTACK AND ATTACK SETUP

We adopt the following attacks toward static GNNs. Specifically, we adopt the attack at the same time as our attack time by attacking the existing dynamic graph as a static graph:

**FakeNode (Wang et al., 2018)** uses a greedy approach to generate edges of malicious nodes and their corresponding features to mislead the static GNN predictions. Note that this approach assumes that the added nodes/edges will be kept in the graph, so we keep the fake nodes and edges still after the attack timestamp. Differently, the attacking nodes in MemFrezzing are removed after the attack.

**TDGIA (Zou et al., 2021)** is a cutting-edge Graph Injection Attack tailored to compromise static GNNs. This method exploits the inherent vulnerabilities of GNNs and the unique topological characteristics of graphs. In our implementation for each target node, we adhere to the established methodology of TDGIA to identify the top 65% susceptible edges, utilizing their specialized scheme for selecting topologically defective edges. These edges are then optimized using gradient descent. Notably, the scale of modifications applied to each target node in the TDGIA method is substantially larger than our approach, involving adjustments to 65% edges per node instead of just one edge per node. Furthermore, these modifications will be kept after the attack instead of being removed as our attack.

**Meta_Attack_Heuristic (Li et al., 2022)** is a heuristic-based attack inspired by the meta attack (Zügner & Günnemann, 2019). This heuristic-based approach is an evolution of the original meta-attack, which relied on gradient-based edge selection. The updated heuristic version demonstrates greater versatility across a variety of GNN models and large-scale graphs, and it exhibits enhanced effectiveness compared to its predecessor. Notably, the meta-attack and its heuristic counterpart operate under the assumption that edges lack attributes. Consequently, in our application, we assign an all-zero feature to the fake edges inserted as part of the attack process.

For all attacks (including our attack), We select ranges of noisy messages (i.e., magnitudes of message features ) between -1 and 1 since -1 and 1 are the theoretical minimum and maximum values of the clean messages. The messages in TGNNs are usually memories of the nodes updated from previous timestamps, which have activation functions such as tanh/cosine functions right before the outputs. Therefore, all features of these messages (i.e., memories) should be within the range of -1 and 1 as the minimum and maximum values of the activation functions (i.e., tanh). Therefore, using -1 and 1 produces messages that are exactly similar to those of the other features in the graph.

All adversarial messages/nodes in the baselines and our attacks use the $\Delta T$ of the most recent clean message on the victim nodes. For example, if we attack node n, whose most recent message before our attack uses $\Delta T_k$ at its updating, then the timestamp of the fake messages added to this node will be $\Delta T_k$ as well. It is also worth mentioning that the delta T has limited effects on the updating process, as we discussed in Appendix A.2.

For all attacks, we define the attack budget as the ratio of nodes that are affected. To ensure a fair comparison, all attacks target the same set of victim nodes (the highest-degree ones). We would also like to mention that, although targeting these high-degree nodes, all benchmarked attacks, including MemFreezing, either inject one-degree nodes or edges into the graph and affect the same number of victim nodes at the time of the attack.

Specifically, MemFreezing targets high-degree nodes by introducing a temporary fake node for each target and creating an event (i.e., an edge) between the fake node and the target. In this way, MemFreezing, like FakeNode, injects nodes with a degree of one into the graph. However, unlike FakeNode, which retains the injected fake nodes and can potentially cause stronger adversarial effects, MemFreezing removes these fake nodes after the attack, minimizing structural changes while inducing long-lasting adversarial effects. Therefore, given a graph with $V$ nodes and $E$ edges and targeting $N = 5\%V$ victim nodes (i.e., 5% budget), MemFreezing adds $N$ fake edges. Since nodes typically have a degree greater than one, $K = 5\%E > 5\%V = N$, the edge changes are less than 5% edges.

## C.3 BASELINE DEFENSES SETUP

We adopt the following defensive strategies for the vanilla TGNN models:

**Adversarial Training:** In line with the approach detailed in (Madry et al., 2017), we introduce perturbations to the node memories in TGNN models during the training. We then employ a minimax adversarial training scheme to enhance the robustness of the TGNN model against these perturbations.

**Regularization under empirical Lipschitz bound:** Following the methodology in (Jia et al., 2023), we minimize the empirical Lipschitz bound during the TGNN training process, where the empirical Lipschitz bound, $L$, is computed by:

$$L = \sup_{\Delta} \frac{||f(x + \Delta) - f(x)||_2^2}{||\Delta||_2^2} \tag{19}$$

This regularization aims to bound the effectiveness of small perturbations, such as adversarial examples.

**GNNGurad:** Following the insights that only the similar node may provide significant information for prediction, GNNGuardZhang & Zitnik (2020) adopts a cosine-similarity-based approach to discount the messages passing between dissimilar nodes.

Notably, most robust GCN models, such as RobustGCN, SGCN, GraphSAGE, and TAGCN mentioned in (Zou et al., 2021), are primarily tailored for static graph benchmarks. Given their design constraints, these models are unsuited for TGNN setup with dynamic graph benchmarks and do not offer a viable defense for the TGNN models targeted by our attack.

## C.4   EXTRA MAIN RESULTS

Here we report edge prediction accuracies on REDDIT-TITLE in Table 4, and node classification AUCs on WIKI in Table 5. The results indicate that: (1) The static attacks cannot last long and affect future nods. (2) Our approach can be more and more effective after the attack time.

### C.4.1   EDGE PREDICTION RESULTS (5% ATTACK BUDGET)

Here, we report edge prediction accuracies on REDDIT-TITLE in Table 4. The results indicate that: (1) The static attacks cannot last long and affect future nods. (2) Our approach can be more and more effective after the attack time.

Table 4: Accumulated accuracy of edge prediction in the vanilla/attacked TGNNs over different timestamps on REDDIT-TITLE; lower matrices indicate more effective attacks.

| Dataset | | REDDIT-TITLE | | | |
|---|---|---|---|---|---|
| Model | | TGN | JODIE | Dyrep | ROLAND |
| Vanilla | | 0.93 | 0.92 | 0.91 | 0.91 |
| $t_0$ | FN | 0.76 | 0.82 | 0.77 | 0.79 |
| | Meta_h | 0.86 | 0.83 | 0.88 | 0.85 |
| | TDGIA | **0.72** | **0.81** | **0.74** | **0.76** |
| | ours | 0.84 | 0.85 | 0.81 | 0.78 |
| $t_{25}$ | FN | 0.9 | 0.86 | 0.89 | 0.88 |
| | Meta_h | 0.89 | 0.86 | 0.9 | 0.87 |
| | TDGIA | 0.89 | 0.85 | 0.89 | 0.88 |
| | ours | **0.81** | **0.84** | **0.76** | **0.80** |
| $t_{50}$ | FN | 0.9 | 0.86 | 0.9 | 0.88 |
| | Meta_h | 0.9 | 0.86 | 0.9 | 0.88 |
| | TDGIA | 0.89 | 0.86 | 0.9 | 0.87 |
| | ours | **0.77** | **0.82** | **0.76** | **0.77** |

### C.4.2   NODE CLASSIFICATION RESULTS (5% ATTACK BUDGET)

Here, we report node classification AUCs on WIKI in Table 5. The results are similar to the edge predictions: Static attacks are good at the first attack time but cannot last long and affect future nods. In contrast, MemFrezzing can be more and more effective after the attack time.

Table 5: The AUC of vanilla/attacked TGNNs on the node classification task; lower matrices indicate more effective attacks.

| Dataset | | WIKI | | | |
|---|---|---|---|---|---|
| Model | | TGN | JODIE | Dyrep | ROLAND |
| Vanilla | | 0.90 | 0.88 | 0.89 | 0.90 |
| t0 | FN | **0.77** | 0.87 | **0.75** | 0.78 |
| | Meta | 0.86 | 0.83 | 0.86 | 0.85 |
| | TDGIA | 0.73 | **0.82** | 0.76 | **0.75** |
| | ours | 0.82 | 0.88 | 0.84 | 0.80 |
| t25 | FN | 0.90 | 0.88 | 0.88 | 0.88 |
| | Meta | 0.89 | 0.87 | 0.88 | 0.89 |
| | TDGIA | 0.88 | 0.87 | 0.88 | 0.89 |
| | ours | **0.82** | **0.85** | **0.81** | **0.79** |
| t50 | FN | 0.90 | 0.88 | 0.88 | 0.90 |
| | Meta | 0.90 | 0.89 | 0.90 | 0.90 |
| | TDGIA | 0.90 | 0.88 | 0.88 | 0.90 |
| | ours | **0.80** | **0.85** | **0.77** | **0.77** |

### C.4.3 RESULTS WITH DIFFERENT BUDGET(1% ATTACK BUDGET)

To more comprehensively show the impact of the attack budget, we include detailed results of baselines' and our attacks' effectiveness under the attack budget as 1%. As shown in Table 6, Table 7, and Table 8, our approach can outperform baselines as well, despite fewer nodes being attacked.

Table 6: Accumulated accuracy of edge prediction in the vanilla/attacked TGNNs over different timestamps on `WIKI` and `REDDIT`; The attack budget is 1% for all attacks; lower matrices indicate more effective attacks.

| Dataset | | WIKI | | | | REDDIT | | | |
|---|---|---|---|---|---|---|---|---|---|
| | **Model** | **TGN** | **JODIE** | **Dyrep** | **ROLAND** | **TGN** | **JODIE** | **Dyrep** | **ROLAND** |
| | Vanilla | 0.93 | 0.87 | 0.86 | 0.94 | 0.97 | 0.98 | 0.96 | 0.95 |
| $t_0$ | FN | 0.89 | 0.83 | 0.82 | 0.85 | 0.93 | 0.93 | 0.92 | 0.85 |
| | Meta | 0.92 | 0.85 | 0.83 | 0.89 | 0.95 | 0.96 | 0.94 | 0.93 |
| | TDGIA | 0.83 | 0.81 | 0.77 | 0.83 | 0.89 | 0.88 | 0.88 | 0.8 |
| | ours | 0.9 | 0.82 | 0.84 | 0.9 | 0.93 | 0.94 | 0.94 | 0.86 |
| $t_{25}$ | FN | 0.92 | 0.87 | 0.85 | 0.94 | 0.97 | 0.97 | 0.96 | 0.95 |
| | Meta | 0.93 | 0.86 | 0.85 | 0.93 | 0.95 | 0.98 | 0.95 | 0.94 |
| | TDGIA | 0.91 | 0.84 | 0.83 | 0.93 | 0.94 | 0.96 | 0.96 | 0.92 |
| | ours | 0.8 | 0.82 | 0.82 | 0.88 | 0.81 | 0.84 | 0.91 | 0.84 |
| $t_{50}$ | FN | 0.94 | 0.87 | 0.86 | 0.94 | 0.97 | 0.97 | 0.96 | 0.95 |
| | Meta | 0.94 | 0.87 | 0.86 | 0.93 | 0.96 | 0.98 | 0.95 | 0.95 |
| | TDGIA | 0.94 | 0.87 | 0.85 | 0.93 | 0.96 | 0.97 | 0.95 | 0.93 |
| | ours | 0.85 | 0.81 | 0.80 | 0.86 | 0.83 | 0.84 | 0.91 | 0.83 |

Table 7: Accumulated accuracy of edge prediction in the vanilla/attacked TGNNs over different timestamps on `REDDIT-BODY` and `REDDIT-TITLE`; The attack budget is 1% for all attacks; lower matrices indicate more effective attacks.

| Dataset | | REDDIT-BODY | | | | REDDIT-TITLE | | | |
|---|---|---|---|---|---|---|---|---|---|
| | **Model** | **TGN** | **JODIE** | **Dyrep** | **ROLAND** | **TGN** | **JODIE** | **Dyrep** | **ROLAND** |
| | **Vanilla** | 0.9 | 0.87 | 0.9 | 0.88 | 0.93 | 0.92 | 0.91 | 0.91 |
| $t_0$ | FN | 0.85 | 0.85 | 0.81 | 0.83 | 0.88 | 0.88 | 0.85 | 0.83 |
| | Meta | 0.87 | 0.85 | 0.87 | 0.86 | 0.92 | 0.89 | 0.89 | 0.9 |
| | TDGIA | 0.81 | 0.83 | 0.79 | 0.78 | 0.85 | 0.87 | 0.85 | 0.83 |
| | ours | 0.87 | 0.85 | 0.85 | 0.82 | 0.88 | 0.9 | 0.86 | 0.85 |
| $t_{25}$ | FN | 0.9 | 0.84 | 0.89 | 0.88 | 0.92 | 0.92 | 0.9 | 0.91 |
| | Meta | 0.9 | 0.87 | 0.9 | 0.88 | 0.93 | 0.93 | 0.91 | 0.91 |
| | TDGIA | 0.88 | 0.86 | 0.9 | 0.87 | 0.92 | 0.92 | 0.9 | 0.91 |
| | ours | 0.84 | 0.86 | 0.8 | 0.82 | 0.85 | 0.88 | 0.81 | 0.86 |
| $t_{50}$ | FN | 0.9 | 0.87 | 0.9 | 0.88 | 0.93 | 0.92 | 0.9 | 0.91 |
| | Meta | 0.9 | 0.88 | 0.9 | 0.88 | 0.93 | 0.93 | 0.9 | 0.91 |
| | TDGIA | 0.89 | 0.87 | 0.9 | 0.87 | 0.93 | 0.91 | 0.9 | 0.9 |
| | ours | 0.79 | 0.85 | 0.77 | 0.83 | 0.8 | 0.83 | 0.82 | 0.83 |

Table 8: The AUC of vanilla/attacked TGNNs on the node classification task under 1% node attacked budget; lower matrices indicate more effective attacks.

| Dataset | | **WIKI** | | | |
|---|---|---|---|---|---|
| | **Model** | **TGN** | **JODIE** | **Dyrep** | **ROLAND** |
| | Vanilla | 0.9 | 0.88 | 0.89 | 0.9 |
| | FN | 0.83 | 0.88 | 0.83 | 0.83 |
| | Meta | 0.87 | 0.85 | 0.88 | 0.88 |
| $t_0$ | TDGIA | 0.81 | 0.85 | 0.83 | 0.8 |
| | ours | 0.86 | 0.88 | 0.86 | 0.85 |
| | FN | 0.89 | 0.88 | 0.89 | 0.9 |
| | Meta | 0.9 | 0.88 | 0.88 | 0.89 |
| $t_{25}$ | TDGIA | 0.9 | 0.87 | 0.89 | 0.89 |
| | ours | 0.82 | 0.85 | 0.81 | 0.81 |
| | FN | 0.9 | 0.87 | 0.89 | 0.9 |
| | Meta | 0.9 | 0.88 | 0.89 | 0.9 |
| $t_{50}$ | TDGIA | 0.9 | 0.87 | 0.89 | 0.89 |
| | ours | 0.82 | 0.88 | 0.79 | 0.82 |

### C.4.4 RESULTS WITH LARGE-SCALE DATASET

To measure our approach on a larger dataset, we select the largest temporal graph dataset on the SNAP dataset collectionLeskovec & Sosič (2016)—`Wiki-Talk-Temporal`Paranjape et al. (2017)—for further analysis. This dataset represents Wikipedia users editing each other's Talk page. A directed edge $(u, v, t)$ means user $u$ edited $v$'s talk page at time $t$. **The graph has 1,140,149 nodes and 7,833,140 collected over 2320 days.**

The dataset has non-attributed edges, so we set them as all zero vectors. Note that we set the memory size to 64 instead of 172 to avoid the Out-Of-Memory issue. Due to the time limit, we train TGN and Roland for ten epochs instead of 20 in our prior experimental settings. The results are shown in Table 9. As we can observe, even for a very large graph with a 1% node budget, our attack shows a similar behavior as our prior results –Our attack is long-lasting and can affect more nodes' predictions in the future.

Table 9: Accumulated accuracy of edge prediction in the vanilla/attacked TGNNs over different timestamps on `Wiki-Talk-Temporal`.

| Dataset | | **Wiki-Talk-Temporal** | | | |
|---|---|---|---|---|---|
| **Attack Budget** | | **1%** | | **5%** | |
| | **Model** | **TGN** | **ROLAND** | **TGN** | **ROLAND** |
| | **Vanilla** | 0.97 | 0.98 | 0.97 | 0.98 |
| $t_0$ | FN | 0.89 | 0.90 | 0.83 | 0.88 |
| | ours | 0.94 | 0.91 | 0.86 | 0.88 |
| $t_{25}$ | FN | 0.98 | 0.97 | 0.97 | 0.96 |
| | ours | 0.92 | 0.90 | 0.82 | 0.89 |
| $t_{50}$ | FN | 0.97 | 0.98 | 0.97 | 0.97 |
| | ours | 0.91 | 0.91 | 0.84 | 0.86 |

## C.5 EXTRA RESULTS ON ATTACKS UNDER DEFENSES

We include the results of two attacks, i.e., FakeNode and MemFrezzing, under the two defenses, i.e., `adv_train` and `Lip_reg`, on two TGNN models, i.e., JODIE and Dyrep. The observations are similar to the prior analysis.

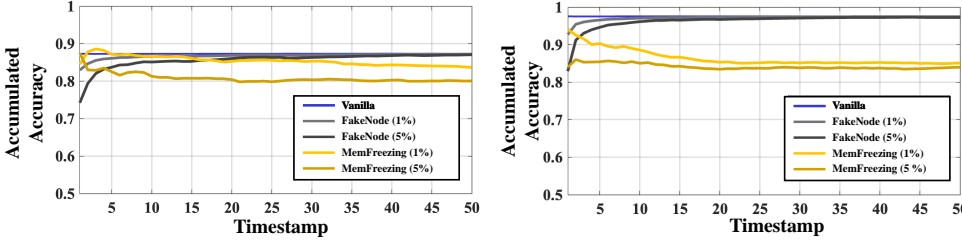

Figure 12: Accumulated accuracies of DyRep under `Adv_train`(left), and `Lip_reg`(right) with FakeNode and our attack on `WIKI`.

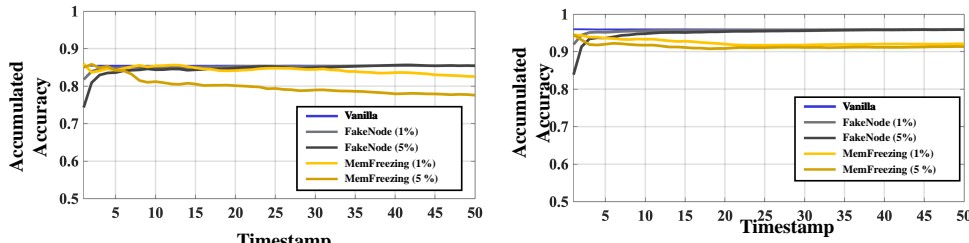

Figure 13: Accumulated accuracies of JODIE under `Adv_train`(left), and `Lip_reg`(right) with FakeNode and our attack on `WIKI`.

To give a more in-depth evaluation, we design a defending method by leveraging the data-filtering concept in GNNGuardZhang & Zitnik (2020) for the evasion attack. Specifically, following the insights that only the similar node may provide significant information for prediction, GNNGuard adopts a cosine-similarity-based approach to discount the messages passing between dissimilar nodes. So, we also use the cosine similarities to rank and filter the messages. Specifically, similar to the GNNGuard, we compute the similarities between two nodes. For each node, we normalize the similarities between it and its neighbors, then prune the lower 50% (same as GNNGuard). We show the experiment results in Table 10.

Table 10: Attack Performance under the **GNNGurad**.

| Attack Budget | | 1% | | | | 5% | | | |
|---|---|---|---|---|---|---|---|---|---|
| Dataset | | WIKI | | REDDIT | | WIKI | | REDDIT | |
| Model | | TGN | ROLAND | TGN | ROLAND | TGN | ROLAND | TGN | ROLAND |
| Vanilla | | 0.93 | 0.94 | 0.96 | 0.95 | 0.93 | 0.94 | 0.96 | 0.95 |
| After defense Acc. | | 0.92 | 0.91 | 0.94 | 0.90 | 0.92 | 0.91 | 0.94 | 0.90 |
| $t_0$ | FN | 0.87 | 0.81 | 0.9 | 0.84 | 0.82 | 0.86 | 0.82 | 0.81 |
| | ours | 0.87 | 0.88 | 0.91 | 0.88 | 0.85 | 0.83 | 0.8 | 0.82 |
| $t_{25}$ | FN | 0.9 | 0.91 | 0.93 | 0.9 | 0.91 | 0.91 | 0.92 | 0.9 |
| | ours | 0.84 | 0.87 | 0.82 | 0.81 | 0.79 | 0.81 | 0.81 | 0.8 |
| $t_{50}$ | FN | 0.92 | 0.91 | 0.94 | 0.9 | 0.92 | 0.91 | 0.93 | 0.9 |
| | ours | 0.83 | 0.85 | 0.81 | 0.81 | 0.76 | 0.82 | 0.8 | 0.83 |

## C.6 Extra results with gradually injected attacks

MemFreezing can be effective in both one-time and multiple-time attacks. We show the results of multiple-time attacks, in which attacks are injected right before $t_0, t_5, t_{10}, t_{15}$ with 1% attack budget (i.e., 1% of all nodes) each time. The results are shown in Table 14.

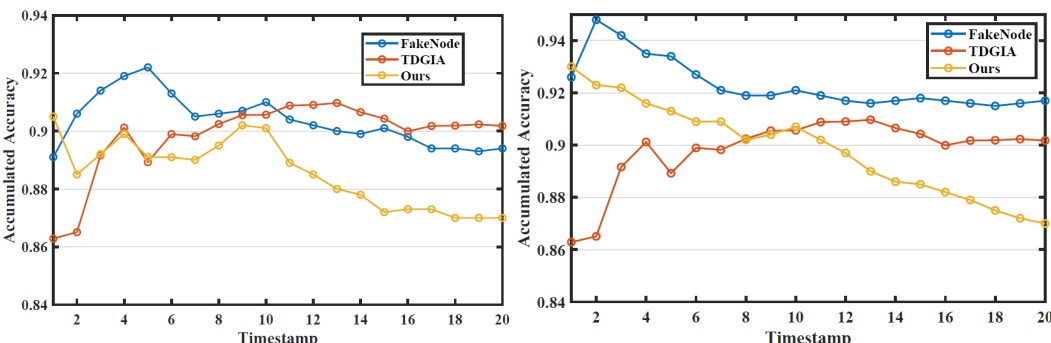

Figure 14: The accumulated accuracy under gradually injected attack to TGN on WIKI(left) and REDDIT(right). The attacks are injected right before $t_0, t_5, t_{10}, t_{15}$ with 1% attack budget (i.e., 1% of all nodes) each time.

As one can observe, with multiple attack times, MemFreezing effectively decreases accuracies, while the FakeNode and TDGIA attacks have shorter effective periods and fail to achieve similar accuracy drops. This is because the attack introduced by these baseline attacks will be weakened once there are changes between the graph at the attack and the prediction timestamp, and even multiple-time attacks cannot ensure that the attacks are just injected right before each prediction; in contrast, the noises from our attack can last over graph changes and even be boosted by future attacks.

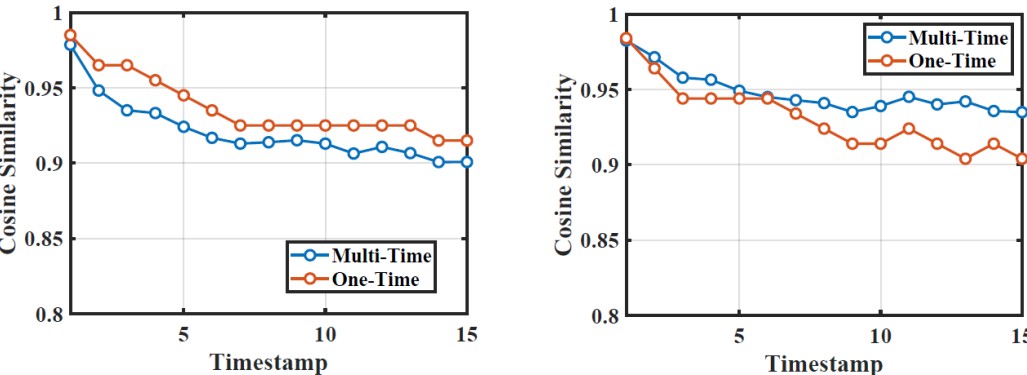

Figure 15: The similarities between victim nodes' initial noisy memories (at the time of the attack) and themselves'/their subsequent neighbors' memories in MemFreezing under one-time attack setup and multiple-times attack setup.

To evaluate the effectiveness of cross-freezing under multiple-time attack cases, we investigate the similarities between victim nodes' initial noisy memories (at the time of the attack) and themselves'/their subsequent neighbors' memories in MemFreezing under one-time attack setup and multiple-times attack setup (following the setup in Figure 7 in our paper). As shown in Figure 15, despite multiple times of injections, MemFreezing significantly raises the similarities between nodes' memories. The results demonstrate that the cross-freezing mechanism works effectively under multiple time attacks.

### C.7 EXTRA RESULTS ON INJECTING ATTACKS AT DIFFERENT TIME STAMPS

To examine if MemFreezing can be effective despite the time of injection. We test its effectiveness under different injection timestamps instead of $t_0$, then evaluate its performance in the subsequent 50 timestamps. For instance, we may inject it at $t_{10}$ and then evaluate the accumulated accuracies in the original TGNN models and those under attack at $t_{50}$. The results of TGN on WIKI and REDDIT are shown in Figure 16. As one can observe, the attack effects remain similar despite its injecting time, demonstrating that MemFreezing can yield long-lasting and contagious attack at arbitrary attack time.

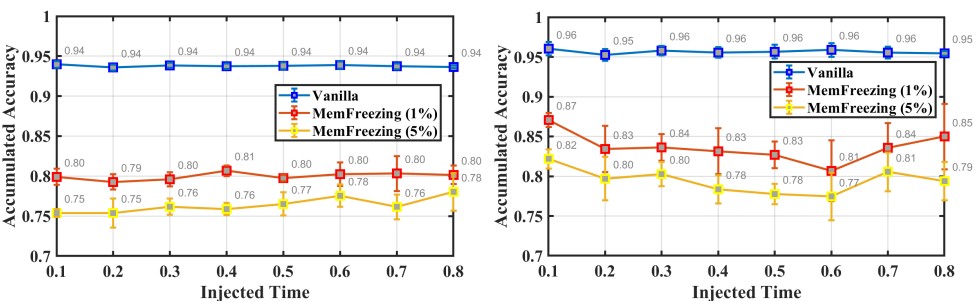

Figure 16: Accumulated accuracy under attack at various timestamps on TGN for WIKI (left) and REDDIT (right). Attacks are injected at 10%, 20%, 30%, 40%, 50%, 60%, 70%, and 80% of the total test set.

### C.8 EXTRA ABLATION STUDY

We include the results for the ablation studies under the TGN model and REDDIT dataset in Figure 17. The results show a similar pattern as we observed in Section 5.

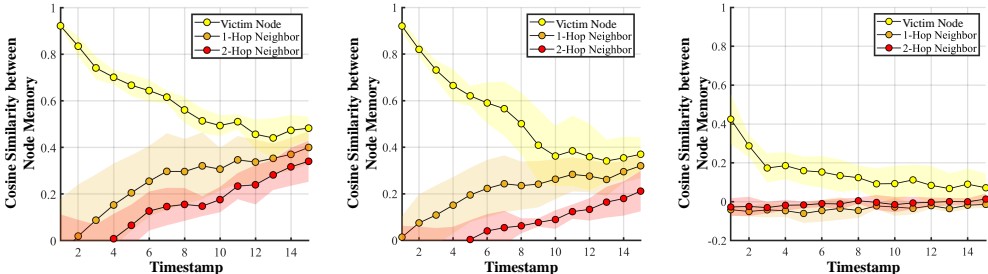

Figure 17: The similarities between victim nodes' initial noisy memories (at the time of the attack) and themselves'/their subsequent neighbors' memories in MemFreezing w/o (left) converge state, MemFreezing w/o freezing loss (middle), and regular nodes (right). All results are from the TGN model and REDDIT dataset.

### C.9 ANALYSIS ON FREEZING OBJECTIVE

To demonstrate the challenge of maximizing prediction losses, we add an extra term, $\mathcal{L}_u^{adv}$ to maximize the loss of predictions. Specifically, for each node $u$ we change Equation 7 in our paper as follows,

$$\mathcal{L}_u = \mathcal{L}_u^{freeze} + \mathcal{L}_u^{prop} - \gamma \cdot \mathcal{L}_u^{adv}$$

We use a coefficient $\gamma$ to control the ratio of adversarial losses. The adversarial loss $\mathcal{L}_u^{adv}$ is defined as follows,

$$\mathcal{L}_u^{adv} = \sum_i \ell(y_i, t_i) \mid i \in N(u)$$

In which $y_i$ presents the prediction result for the node $i$, $t_i$ is the ground truth of the prediction, and $\ell(y_i, t_i)$ indicates the binary-cross-entropy loss between them. Similar to baselines, for each node $u$, the objection function is to maximize the prediction loss of all its neighbors. We present the prediction accuracies under different $\gamma$ selections in Figure 18.

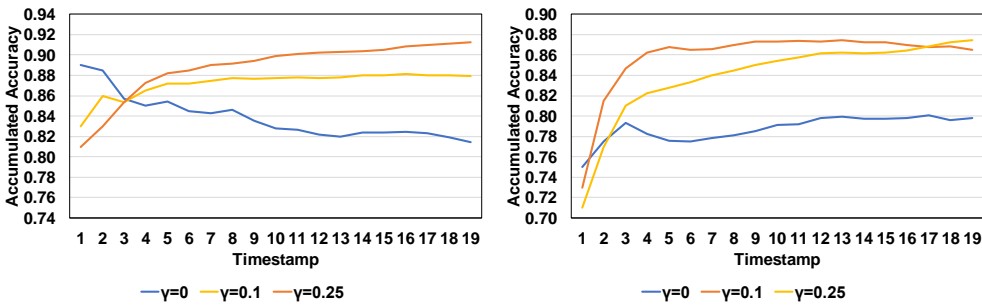

Figure 18: The accumulated accuracy with maximizing prediction losses under different $\gamma$ selections on TGN in WIKI(left) and REDDIT(right).

As shown in the figure, maximizing the adversarial losses can harm the predictions in the first batch (if and only if the predictions are made immediately after the attack). In the later batches, the effectiveness of the noise decreases drastically.

To further understand the reasons behind this, we investigate the similarities between victim nodes' initial noisy memories (at the time of the attack) and their memories in the future—termed as Persist Similarity—in Figure 19, the similarities between victim nodes' memories and their neighbors' memories—called Propagate Similarity in Figure 20.

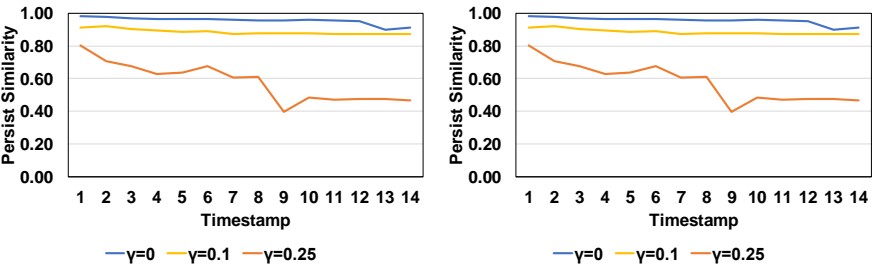

Figure 19: The similarities between victim nodes' initial noisy memories (at the time of the attack) and their memories in the future. The results are collected from TGN on WIKI(left) and REDDIT(right).

As one can observe, while introducing the adversarial losses, both persist and propagate similarities drop significantly, indicating that the nodes' memories cannot maintain the noisy states and may recover soon.

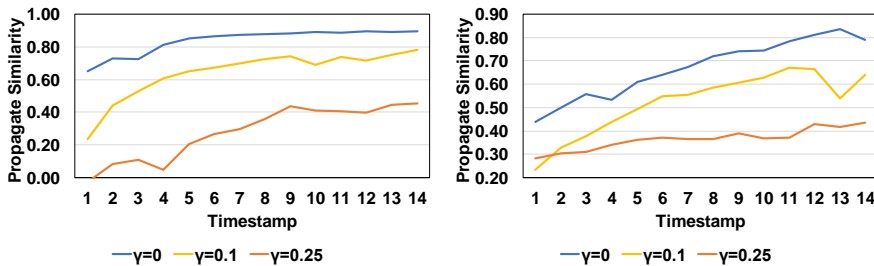

Figure 20: The similarities between victim nodes' memories and their future neighbors' memories. The results are collected from TGN on `WIKI`(left) and `REDDIT`(right).

## C.10 STEALTHNESS ANALYSIS

As discussed in Appendix C.2, we select ranges of noisy messages between -1 and 1 since -1 and 1 are the theoretical minimum and maximum values of the clean messages. To further investigate if the MemFreezing attack introduces enough stealth fake events/nodes, we further investigate the range of message-wise means (i.e., means of all features over each message) and message-wise standard deviation (i.e., the standard deviation of all features over each message) for clean and noisy messages produced by different attacks in Table 11.

Table 11: Ranges of message-wise mean and standard deviation over all of the clean messages (Clean) and noisy messages produced by MemFreezing in `WIKI` and `REDDIT`.

| | WIKI | | REDDIT | |
|---|---|---|---|---|
| | Mean[min,max] | Std[min,max] | Mean[min,max] | Std[min,max] |
| Clean | [-0.033, 0.106] | [0.206, 0.866] | [-0.093, 0.146] | [0.202, 0.789] |
| MemFreezing | [-0.014, 0.044] | [0.426, 0.570] | [-0.012, 0.038] | [0.580, 0.695] |
| FakeNode | [0.003 , 0.008 ] | [0.628 , 0.702] | [-0.030, 0.018] | [0.525, 0.686] |

The range of mean and std of our noisy messages are included within the range of those in the clean message and are similar to the baseline attack, demonstrating that their distributions or magnitudes are similar to the other features in the graph. Moreover, MemFreezing can effectively penetrate the defenses of GNNGuard, which uses similarity to filter susceptible messages in which the nodes/events with apparently different information (i.e., having low similarities compared to other nodes/events), as shown in Appendix C.5. In summary, the results indicate that MemFreezing can freeze node memories in TGNN without introducing significant different nodes/events that can be detected by existing GNN adversarial defenses.

## C.11 EXTRA SENSITIVITY STUDY

We include more results for different target node sampling strategies and attack budgets in Figure 21. The results show a similar pattern as we observed in Section 5.

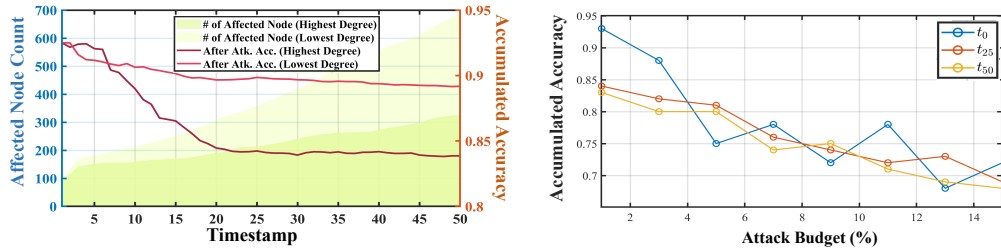

Figure 21: (left) Comparison between two strategies for selecting the injected node: lowest degree and highest degree nodes. Count of affected nodes and overall accuracy over time. (RIGHT) The accumulated accuracy at $t_0$, $t_{25}$, and $t_{50}$ under different attack budgets (% of total nodes). All results above are from TGN and REDDIT

## C.12    ACCUMULATED ACCURACIES OVER TIME ON DIVERSE MODELS

We report the accumulated accuracies over time collected from TGN, JODIE, and Dyrep on the WIKI and REDDIT datasets. The results include model accuracies under the vanilla (i.e., un-attacked), baseline (i.e., FakeNode), and our (i.e., MemFrezzing) attacks in edge prediction tasks.

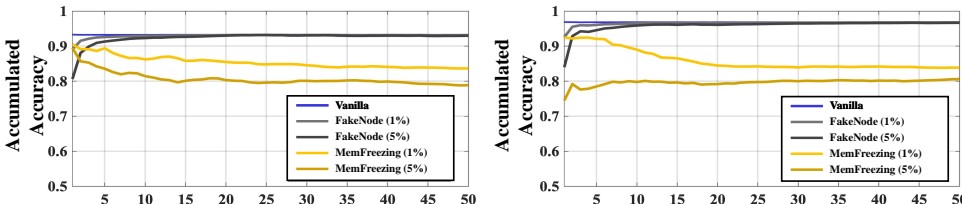

Figure 22: Accumulated accuracies of TGN under different attacks in link prediction tasks over time in WIKI (left) and REDDIT (right) datasets.

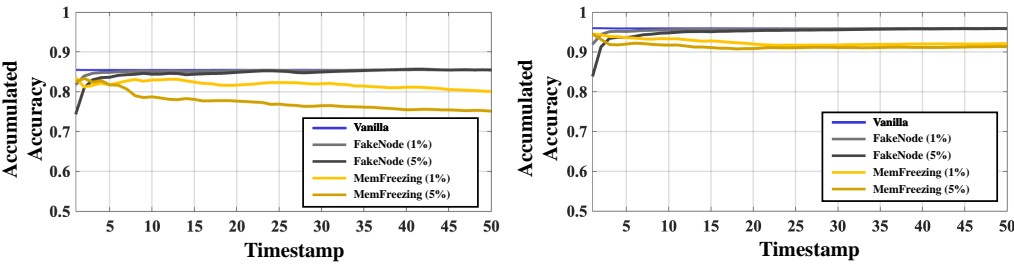

Figure 23: Accumulated accuracies of JODIE under different attacks in link predictions over time with WIKI (left) and REDDIT (right) datasets.

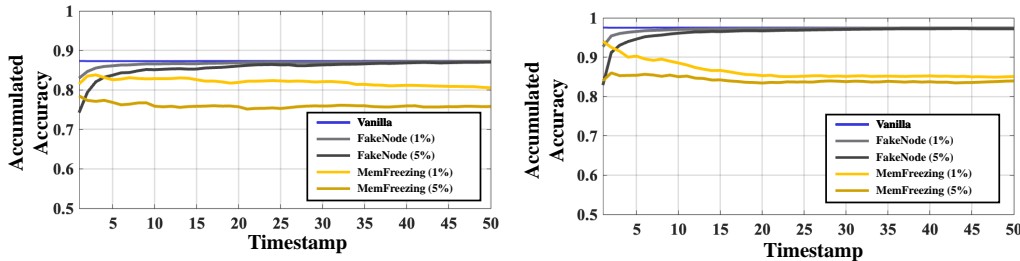

Figure 24: Accumulated accuracies of Dyrep under different attacks in link predictions over time with WIKI (left) and REDDIT (right) datasets.

## C.13 AFFECTED NODES

We report the number and accumulated accuracies over time of affected nodes over time in JODIE and Dyrep on the WIKI and REDDIT datasets. The results include model accuracies under our (i.e., MemFrezzing) attack in edge prediction tasks.

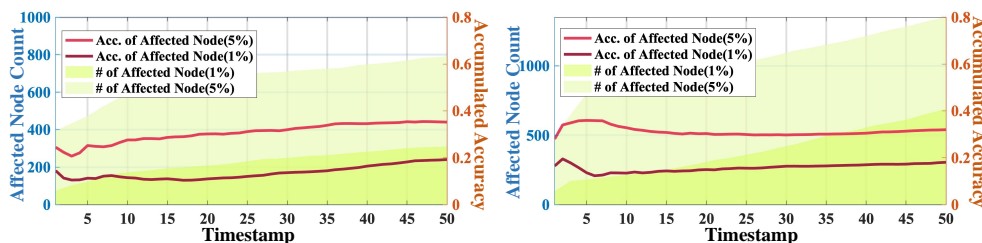

Figure 25: Count of affected nodes (presented as the colored areas) and their accumulated accuracies (presented as lines) in WIKI (left) and REDDIT (right) over time. The data are collected in TGN.

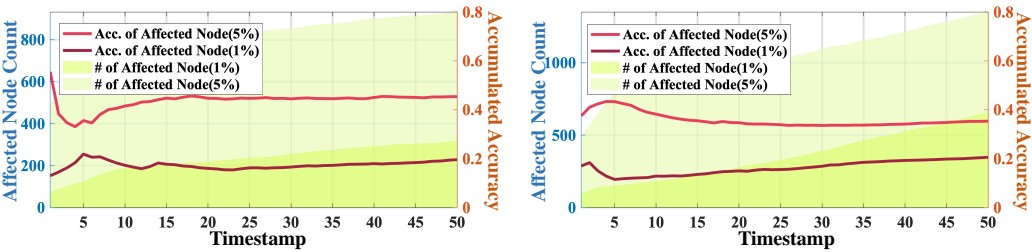

Figure 26: Count of affected nodes (presented as the colored areas) and their accumulated accuracies (presented as lines) in WIKI (left) and REDDIT (right) over time. The data are collected in JODIE.

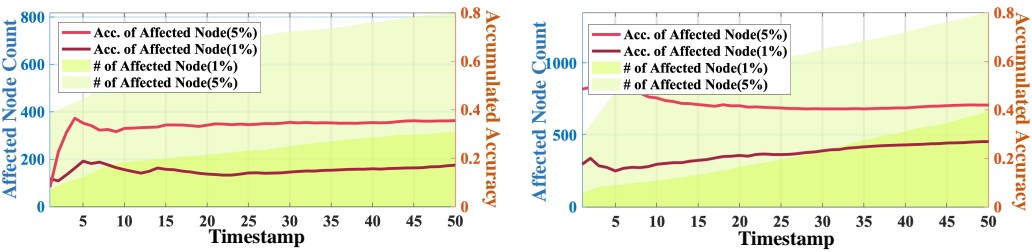

Figure 27: Count of affected nodes (presented as the colored areas) and their accumulated accuracies (presented as lines) in WIKI (left) and REDDIT (right) over time. The data are collected in Dyrep.

## C.14 NOISE PROPAGATING

We report the cosine similarities between the initial victim node and its neighbors over time in JODIE and Dyrep on the `WIKI` and `REDDIT` datasets. The results include similarities under our (i.e., MemFrezzing) attack in edge prediction tasks.

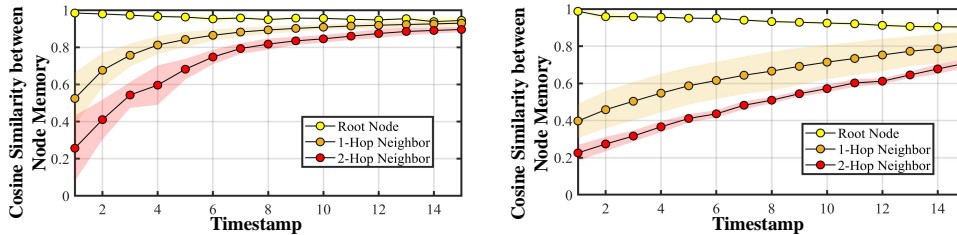

Figure 28: The similarities between victim nodes' initial noisy memories (at the time of the attack) and themselves'/their subsequent neighbors' memories in `WIKI` (left) and `REDDIT` (right) over time. The data are collected in TGN.

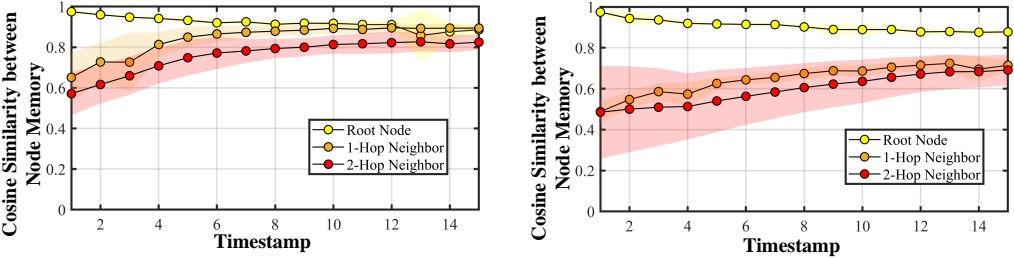

Figure 29: The cosine similarities between victim nodes' initial memory (at the time of the attack) and themselves/their subsequent neighbors' memories in `WIKI` (left) and `REDDIT` (right) over time. The data are collected in JODIE.

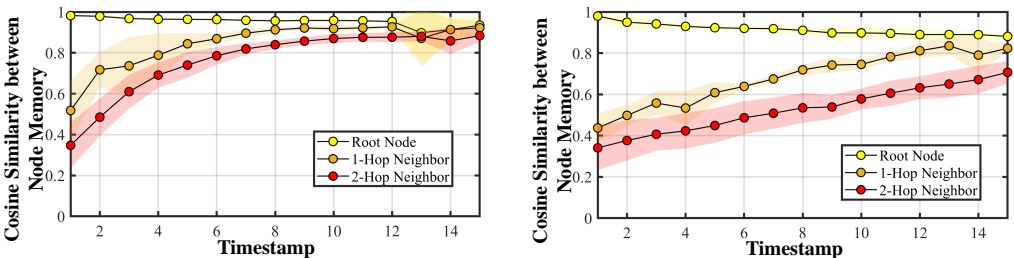

Figure 30: The cosine similarities between victim nodes' initial memory (at the time of the attack) and themselves/their subsequent neighbors' memories in `WIKI` (left) and `REDDIT` (right) over time. The data are collected in Dyrep.

## C.15 ANALYSIS ON FUTURE SIMULATION

To further understand if using nodes' current neighbor can be effective in extremely irregular and random graphs, we conduct the following experiments.

First, we further analyze the similarity among nodes' neighbors in diverse datasets using the same setup (e.g., model) as Figure 4(d). As shown in Figure 31, generally, nodes tend to have similar neighbors across diverse datasets. Hence, using current neighbors reasonably approximates future graph changes in practice.

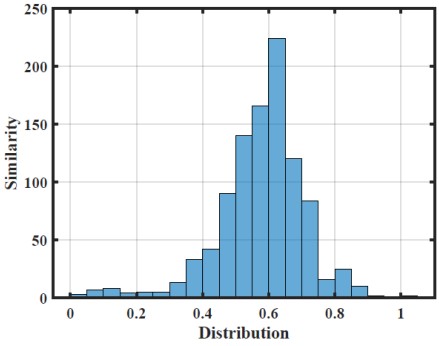 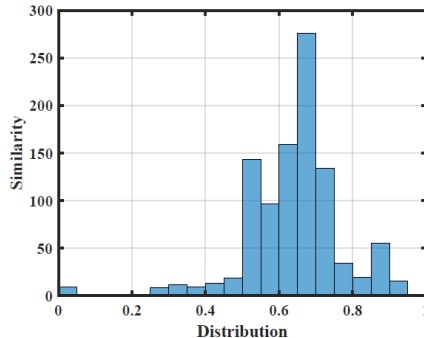

Figure 31: The distribution of cosine similarities among the ideal frozen states in different nodes in REDDIT and REDDIT-BODY datasets.

To investigate if our future neighbor simulation scheme is sufficient to freeze neighbors under irregular or highly random dynamic graphs, we simulate an irregular and random graph on top of the Wikipedia dataset. Specifically, we have victim nodes in the graph connected to nodes with random memories in the future timestamps. We also explored an alternative scheme to investigate whether the heuristic could be further enhanced. Specifically, in this alternative, we simulate nodes' future neighbors using nodes with random memories.

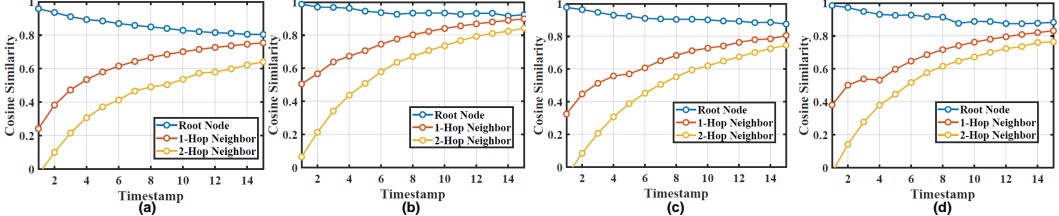

Figure 32: The similarities between victim nodes' initial noisy memories (at the time of the attack) and themselves'/their subsequent neighbors' memories in WIKI dataset and its randomized version under vanilla cases including (a) Using current neighbor for simulation under a noisy future, (b) Using current neighbor for simulation under a normal future, (c) Using random memory neighbor for simulation under a noisy future, (d) Using random memory neighbor for simulation under a normal future.

As shown in Figure 32(a), although resulting in lower similarities, MemFreezing effectively freezes these random neighbors (as shown in (a)). This demonstrates that our future simulation schemes are effective under even (i.e., Current Simulation) in irregular setups. The reason behind this is that, in addition to using current neighbors, we also simulate "new future neighbors" with all-zero memories, which further enhance the noise's capability to freeze unseen nodes.

Although the alternative scheme (i.e., Random Simulation) performs better under random neighbor cases (i.e., Noise Future), as shown in Figure 32(c); it shows worse performances in the real cases (i.e., Normal Future), as shown in Figure 32(d) compared to as shown in Figure 32(b). These findings collectively suggest that using current neighbors as surrogates is both practical and effective, even in challenging dynamic graph scenarios.

## C.16   EFFECTIVE IN LSTM-BASED TGNNS

While existing TGNN uses RNN and GRU for node memory updating Rossi et al. (2020); Trivedi et al. (2019); Kumar et al. (2019); You et al. (2022), it is valuable to understand how nodes' memory is frozen under a memory updater with different RNN-variant.

To evaluate the effectiveness of MemFreezing when using LSTM as the memory updater, we replaced the GRU and RNN components in TGN Rossi et al. (2020) with LSTM. We then assessed the performance of MemFreezing and baseline attacks under this new configuration. It is worth mentioning that since LSTM has two memories (i.e., long and short terms), they are different from GRU and RNN used in existing TGNNs. To adapt these two memories into one node memory under existing TGNN frameworks, we concatenate the two memories of a node together as its memory and freeze them altogether.

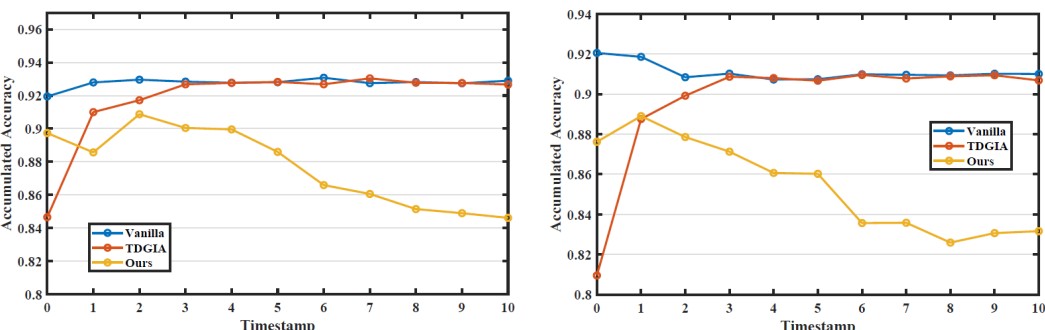

Figure 33: The accumulated accuracy of LSTM-based TGN under no-attack, TDGIA, and Mem-Freezing on `WIKI` (left) and `REDDIT` (right) datasets.

We first investigate the resulting accumulated accuracies in TGN. As shown in Figure 33, the LSTM-based TGN shows better robustness against MemFreezing. However, MemFreezing still effectively compromises predictions of LSTM-based TGN, leading to an average of 8% accuracy drops at $t_{100}$. In contrast, the baseline (i.e., TDGIA) still fails to disturb the predictions under limited-knowledge setups.

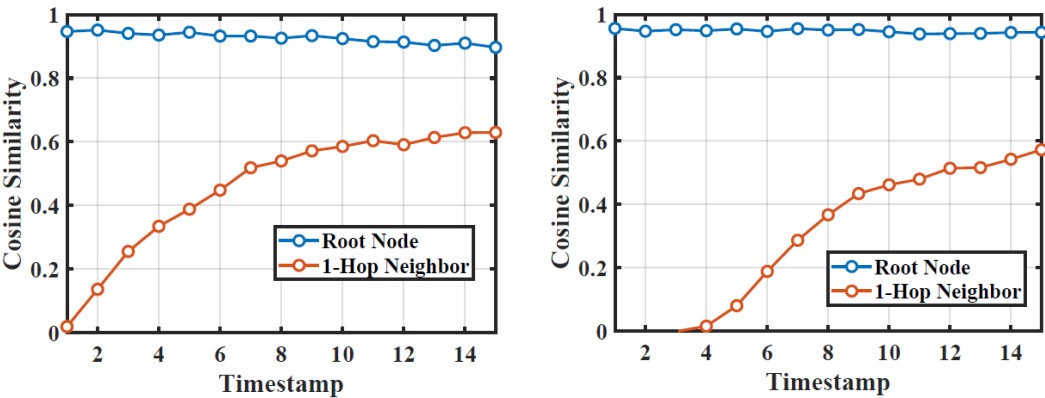

Figure 34: The similarities between victim nodes' initial noisy memories (at the time of the attack) and themselves'/their subsequent neighbors' memories in LSTM-based TGN on the `WIKI` dataset.

The LSTM-based TGN makes it more challenging since the attack has to freeze both long-term and short-term memories. To understand the phenomenon, we further investigate the similarities between the victim nodes' initial memory and its subsequent and 1-hop neighbors' memories. As shown in Figure 34., the similarities between the victim nodes and their 1-hop neighbors are as low as around 0.6, which is not as high as the cases with GRU/RNNs (e.g., over 0.8).

# D DISCUSSION AND FUTURE WORK

## D.1 LIMITS UNDER DIFFERENT MODELS AND GRAPHS.

While the experiment results in Appendix C.4 and Appendix C.11 demonstrated that MemFrezzing can be well-generalized on various inputs, several limitations can be observed according to the performance variance between different models. While our approach can effectively mislead TGN, ROLAND, and DyRep, its effectiveness is less significant on JODIE, which uses differences between a node's current and its last update time to decay the memory. From these observations, we deduce that our attack may encounter limitations in two specific scenarios:

- *Limited Influence of Node Memory on Predictions:* Our attack's effectiveness may be mitigated in situations where the node memory has a relatively minor role in influencing the model's predictions.
- *Usage of Additional Information in TGNN Models*: The effectiveness may also be constrained when the targeted TGNN model incorporates additional information beyond the node memory for its predictive processes.

While our attack strategy outperforms the baselines, these insights highlight potential limitations under certain model-specific conditions.

## D.2 POTENTIAL DEFENSES.

While we demonstrate that many existing defense schemes, such as adversarial training or regularization, are less effective on our attacks, we expect a potential attack-oriented defense scheme for our attack using memory filtering. Specifically, a potential defensive approach for our attack is to pay less attention to the nodes' memory and rely more on their current input adaptively.

This scheme stems from the observation that our attacks are less effective on JODIE in node classification tasks. One key difference in JODIE is that it decays the node memory based on the time differences between the prediction time and the node's last update time. This mechanism introduces more hints (i.e., time differences) in addition to the memory itself, which cannot be effectively distorted by the attacks and yields some crucial information. For example, a Wikipedia user is less likely to be banned if he/she makes a new post after being inactive for a long while.

Therefore, using this non-memory information or current information that does not interact with node memory could effectively hinder adversarial noises. To this end, an intelligent defense mechanism can judiciously filter out the memory and adaptively focus more on non-memory information if the memory is suspicious or potentially noisy.

# E    COMPLEXITY OF THE MEMFREEZING ATTACK

We further approximate the time complexity of the MemFreezing, as it is crucial to understand its practicality. The time complexity of MemFreezing is approximately $\mathcal{O}(V + VD)$, where $V$ is the number of victim nodes being attacked and $D$ is their average degree.

The computation of MemFreezing can be divided into three main parts:

1. **Finding the Stable State**: For each victim node in $V$, we iteratively update its state using its two support neighbors until reaching the ideal stable state. Assuming a constant number of iterations for convergence, this step incurs a time complexity of $\mathcal{O}(V)$.

2. **Solving the Target Memory Using SGD**: For each victim node, we optimize the target memory state using stochastic gradient descent (SGD), considering (a) The node itself, (b) Its two support neighbors and (c)Its augmented neighbors. The total set has a size of at most $D+20$ (current neighbors plus simulated neighbors), where $D$ approximates the number of the node's current neighbors. This optimization incurs a cost of $\mathcal{O}(D)$ per node, leading to a total time complexity of $\mathcal{O}(VD)$ across $V$ victim nodes with $D$ average degree.

3. **Introducing Fake Neighbors**: For each victim node, we compute and inject a fake neighbor to introduce noise. This step has a cost of $\mathcal{O}(1)$ per node, resulting in $\mathcal{O}(V)$ overall.

In summary, the overall time complexity of MemFreezing is dominated by the SGD optimization step for getting noisy memory, resulting in $\mathcal{O}(V + VD)$ time complexity. Under the worst cases, in which $D = V$ (e.g., fully connected graph), the complexity is $\mathcal{O}(V^2)$

