# OpenReview forum: "MEMFREEZING: TOWARDS PRACTICAL ADVERSARIAL ATTACKS ON TEMPORAL GRAPH NEURAL NETWORKS"
_ICLR.cc/2025/Conference — Submitted to ICLR 2025_

### Official Review · Reviewer_QqgZ · 2024-11-02

**Soundness:** 3
**Presentation:** 3
**Contribution:** 3
**Rating:** 8
**Confidence:** 4

**Summary:**

This work represents MemFreezing, an adversarial attack framework for a Temporal Graph Neural Network(TGNN) that targets the intrinsic node memory mechanism. MemFreezing disables node memory in TGNNs by forcing them into stable states, achieved by a novel cross-freezing mechanism and future simulation. Empirical results have demonstrated that the proposed attack causes a long-lasting frozen state of affected nodes and can spread this impact to current and future neighbour nodes.

**Strengths:**

- The paper proposed novel adversarial attack strategies on TGNNs by focusing on the memory component of TGNN models
- MemFreezing can persist impact on TGNNs performance over timestamps by effectively freezing the memory of the affected node.
- MemFreezing disturbs the prediction of TGNNs not only on affected nodes but also on future nodes of the networks.
- The paper pointed out the impractical strategies of existing adversarial attacks on temporal graphs and introduced more effective adversarial strategies with limit k, MemFreezing, under practical setups
- The effects of MemFreezing are evaluated from different vanilla TGNNs to defence models, from small to large scale datasets.

**Weaknesses:**

- Notation. Consistency in notions could enhance the readability of work. This work uses different notation to indicate nodes (i.e. $node_1$, $u$). In addition, in equation (1), x(t_1) is considered as a set of events at timestamp $t_1$, and the $x(t)$ at line 136 indicates an event. Different notations are needed to differentiate a set of events from a single event to avoid confusion.
- Typos. Line 480 mentions an unclear definition of “one-hot” neighbour. “MemFreezing (1%)” is repeated in Figures 12 and 13.
- This adversarial attack strategy only works with node-memory-based TGNNs, limiting MemFreezing's contribution to evaluating the robustness of other TGNNs, such as EdgeBank[1]. But the authors have acknowledged this issue.

**Questions:**

- What is the time complexity of MemFreezing?
- Empirical experiments have shown that MemFreezing can persistently decrease the TGNN models. However, JODIE[2] and Dyrep[3] use RNN to maintain node memory, while TGN[4] and ROLAND[5] adopt GRU. How does MemFreezing perform on other variants of RNNs, especially LSTM?

[1] Poursafaei, Farimah, et al. "Towards better evaluation for dynamic link prediction." Advances in Neural Information Processing Systems 35 (2022): 32928-32941
[2] Kumar, Srijan, Xikun Zhang, and Jure Leskovec. "Predicting dynamic embedding trajectory in temporal interaction networks." Proceedings of the 25th ACM SIGKDD international conference on knowledge discovery & data mining. 2019

[3] Trivedi, Rakshit, et al. "Dyrep: Learning representations over dynamic graphs." International conference on learning representations. 2019.

[4] Rossi, Emanuele, et al. "Temporal graph networks for deep learning on dynamic graphs. arXiv 2020." arXiv preprint arXiv:2006.10637.

[5] You, Jiaxuan, Tianyu Du, and Jure Leskovec. "ROLAND: graph learning framework for dynamic graphs." Proceedings of the 28th ACM SIGKDD conference on knowledge discovery and data mining. 2022.

* None of these are our articles.

---

> ### Author Response · Authors · 2024-11-24
> **Author Response to Reviewer QqgZ (1/2)**
>
> **We sincerely thank the reviewer for the positive feedback and valuable comments. In response, we analyze the time complexity of our attack and add results on LSTM-based TGNNs. We also revise our paper following the suggestions from the reviewer. We hope our response can help clarify the reviewer's questions.**
>
> ---
> ### **Q1. What is the time complexity of MemFreezing?**
>
> Thank you for the valuable question. Studying the time complexity of an attack is crucial for understanding its practicality. The time complexity of MemFreezing is approximately $O(V + VD)$, where $V$ is the number of victim nodes being attacked and $D$ is their average degree.
>
> The computation can be divided into three main parts:
>
> 1. **Finding the Stable State**: For each victim node in a total of $V$ nodes, we iteratively update its state using its two support neighbors until reaching the ideal stable state. Assuming a constant number of iterations for convergence, this step incurs a time complexity of $\mathcal{O}(V)$.
>
> 2. **Solving the Target Memory Using SGD**: For each victim node, we optimize the target memory state using stochastic gradient descent (SGD), considering (a) the node itself, (b) its two support neighbors, and (3) its augmented neighbor, with the total set of size less than $D + 20$ (current neighbors + those simulated neighbors), where $D$ approximate the amount of node’s current neighbors. This optimization incurs an $\mathcal{O}(D)$ cost per node, leading to a total time complexity of $\mathcal{O}(VD)$ across V victim nodes with D average degree.
>
> 3. **Introducing Fake Neighbors**: For each victim node, we compute and inject a fake neighbor to introduce noise. This step has a cost of O(1) per node, resulting in overall $\mathcal{O}(V)$ time complexity.
>
> In summary, the overall time complexity of MemFreezing is dominated by the SGD optimization step for getting noisy memory, resulting in $\mathcal{O}(V + VD)$ time complexity. Under the worst cases, in which $D=V$ (e.g., fully connected graph), the complexity is $\mathcal{O}(V^2)$. We also added the above analysis to our revised paper as Appendix E.

---

> ### Author Response · Authors · 2024-11-24
> **Author Response to Reviewer QqgZ (2/2)**
>
> ---
> ### **Q2. What if the memory updater in TGNNs uses LSTM?**
> Thank you for the thoughtful question. It is valuable to understand how nodes’ memory is frozen under a memory updater with different RNN-variant.
>
> To evaluate the effectiveness of MemFreezing when using LSTM as the memory updater, we replaced the GRU and RNN components in TGN with LSTM. We then assessed the performance of MemFreezing and baseline attacks under this new configuration. It is worth mentioning that since LSTM has two memories (i.e., long and short terms), they are different from GRU and RNN used in existing TGNNs. To adapt these two memories into one node memory under existing TGNN frameworks, we concatenate the two memories of a node together as its memory and freeze them altogether.
>
> We first investigate the resulting accumulated accuracies in TGN. As shown in Table R.6, the LSTM-based TGN shows better robustness against MemFreezing. However, MemFreezing still effectively compromises predictions of LSTM-based TGN, leading to an average of 8% accuracy drops at $t_{100}$. In contrast, the baseline (i.e., TDGIA) still fails to disturb the predictions under limited-knowledge setups.
>
> >**Table R.6. The Accumulated accuracy(accumulated) of LSTM-based TGNN w/o attack (i.e., vanilla), under TDGIA attack and under MemFreezing attack.**
> |        |         |  $t_0$   | $t_{10}$  | $t_{20}$  | $t_{30}$  | $t_{40}$  | $t_{50}$  | $t_{60}$  | $t_{70}$  | $t_{80}$  | $t_{90}$  | $t_{100}$ |
> |--------|---------|------|------|------|------|------|------|------|------|------|------|------|
> | Wiki   | Vanilla | 0.92 | 0.93 | 0.93 | 0.93 | 0.93 | 0.93 | 0.93 | 0.93 | 0.93 | 0.93 | 0.93 |
> |        | TDGIA   | 0.85 | 0.91 | 0.92 | 0.93 | 0.93 | 0.93 | 0.93 | 0.93 | 0.93 | 0.93 | 0.93 |
> |        | Ours    | 0.9  | 0.89 | 0.91 | 0.9  | 0.9  | 0.89 | 0.87 | 0.86 | 0.85 | 0.85 | 0.85 |
> | Reddit | Vanilla | 0.92 | 0.92 | 0.91 | 0.91 | 0.91 | 0.91 | 0.91 | 0.91 | 0.91 | 0.91 | 0.91 |
> |        | TDGIA   | 0.81 | 0.89 | 0.9  | 0.91 | 0.91 | 0.91 | 0.91 | 0.91 | 0.91 | 0.91 | 0.91 |
> |        | Ours    | 0.88 | 0.89 | 0.88 | 0.87 | 0.86 | 0.86 | 0.84 | 0.84 | 0.83 | 0.83 | 0.83 |
>
>
> The LSTM-based TGN makes it more challenging since the attack has to freeze both long-term and short-term memories. To understand the phenomenon, we further investigate the similarities between the victim nodes’ initial memory and its subsequent and 1-hop neighbors’ memories. As shown in Table R.7, the similarities between the victim nodes and their 1-hop neighbors are as low as around 0.6, which is not as high as the cases with GRU/RNNs (e.g., over 0.8).
>
> >**Table R.7. The similarities between victim nodes’ initial noisy memories (at the time of the attack) and themselves’/their subsequent neighbors’ memories in LSTM-based TGN on the Wikipedia dataset.**
> |       |       | $t_1$    | $t_2$   | $t_3$   | $t_4$   | $t_5$   | $t_6$   | $t_7$   | $t_8$   | $t_9$   | $t_{10}$  | $t_{11}$  | $t_{12}$  | $t_{13}$  | $t_{14}$  | $t_{15}$  |
> |--------|-------|-------|-------|------|------|------|------|------|------|------|------|------|------|------|------|------|
> | Wiki   | Root  | 0.95  | 0.95  | 0.94 | 0.94 | 0.94 | 0.93 | 0.93 | 0.93 | 0.93 | 0.92 | 0.91 | 0.91 | 0.90 | 0.91 | 0.90 |
> |        | Hop-1 | 0.02  | 0.14  | 0.26 | 0.33 | 0.39 | 0.45 | 0.52 | 0.54 | 0.57 | 0.59 | 0.60 | 0.59 | 0.61 | 0.63 | 0.63 |
> | Reddit | Root  | 0.96  | 0.95  | 0.95 | 0.95 | 0.95 | 0.95 | 0.95 | 0.95 | 0.95 | 0.94 | 0.94 | 0.94 | 0.94 | 0.94 | 0.94 |
> |        | Hop-1 | -0.02 | -0.03 | 0.00 | 0.02 | 0.08 | 0.19 | 0.29 | 0.37 | 0.43 | 0.46 | 0.48 | 0.51 | 0.52 | 0.54 | 0.57 |
>
> We also added the above analysis to Appendix C.16. in our revised paper.
>
> ---
> ### **Q3. Notation consistency and Typos.**
>
> We appreciate the suggestions from the reviewer and carefully revised our paper, including the following changes:
> >- As for the $node_u$ and $u$, we consistently used its index $u$ to represent a node.
> >- For $x(t_1)$ in equation(1), it also represents a single event, as we used in the following equations. The description is included in our original submitted paper and highlighted in teal color. To address the confusion, we change $x(t)$ in the following statements to $x(t_i)
> >- We correct the typos in line 480 and the legends in Figures 12 and 13.

---

> > ### Comment · Reviewer_QqgZ · 2024-11-25
> >
> > Thank you for your responses.

---

> > > ### Author Response · Authors · 2024-11-25
> > > **Author Response to Reviewer QqgZ**
> > >
> > > We would like to thank you again for your time and your positive comments. It is a great confirmation of our work.

---

### Official Review · Reviewer_YKJ1 · 2024-11-04

**Soundness:** 3
**Presentation:** 1
**Contribution:** 2
**Rating:** 3
**Confidence:** 5

**Summary:**

This paper makes a new definition on practical adversarial attack against temporal graph neural networks, and proposes an event attack method by injecting nodes and edges. The injected edges and nodes are designed to simulate a fake future neighborhood and further to make the nodes memory unchanged in the future updating steps.

**Strengths:**

1. The paper first explains the motivation of “memory freezing” and shows its effectiveness by a preliminary experiment, which offers an interesting insight on TGNN model robustness.

2. Extensive experiments are done to validate the effectiveness of the proposed method.

**Weaknesses:**

1. This attack is limited to a white-box attack, which seems to primarily disobey the target of “practical attack”.

2. The definitions on the attack capability seems not reasonable. In a practical situation, it seems the attacker can inject several timestamps instead of a specific timestamp. Besides, in the experiments only the attack budget on the amount of injected nodes is discussed. Since an injected node may occupying multiple injected edges to different nodes, and therefore a budget on edges amount may be also needed.

3. The writing of the paper is confusing. In the section 3-4 the authors use too much verbal description on the attack problem and the proposed method, while too few notations or definition in formal expressions is given, which makes the paper hard to follow.

**Questions:**

I’m not fully convinced by the setting of attacking on a specific timestamp of the temporal graph, instead of attacking several timestamps. In my assumption, even with the limited “current knowledge” when attacking a specific timestamp, the attacker could still attack a few timestamps by taking a “currently optimized” attack in each step. Is there any special reason or assumed scenarios for this setting of limitation?

---

> ### Author Response · Authors · 2024-11-24
> **Author Response to Reviewer YKJ1 (1/3)**
>
> **We sincerely appreciate the valuable comments and insights from the reviewer. In response, we answer the reviewer’s questions and revised the paper accordingly, including clarifying the rationale of using a white-box setup, providing more results on the multiple-time attacks, clarifying the edge changes in the attacks, and revising the paper for better presentation. We hope our response and revisions can help alleviate the reviewer's concern.**
>
> ---
> ###  **Q1.Will the white-box setup limit the practicality of MemFreezing?**
>
> Thank you for the thoughtful question. We recognize the concern regarding the practicality of white-box attacks.
>
> In terms of model parameters, many TGNN architectures and pre-trained models are open-sourced, making them readily available to adversaries. Furthermore, methods such as insider threats or model extracting [1][2] can be employed to extract model parameters when the model itself is not publicly available. These factors collectively make the white-box attack setup relatively more practical and realistic in many real-world scenarios.
>
> However, future knowledge (e.g., data or updates occurring after $t_0$​) is inherently harder to access due to its temporal and evolving nature. Current methods offer no feasible way to reliably predict future changes in graph structure or labels. As such, we focus on the more challenging constraint of using only past knowledge for attacks while adhering to the white-box setup for model parameters.
> We clarify this point further in Section 3.1 of the revised paper to address this concern explicitly.
>
> [1] Oliynyk, Daryna, Rudolf Mayer, and Andreas Rauber. "I know what you trained last summer: A survey on stealing machine learning models and defences." ACM Computing Surveys 55.14s (2023): 1-41.
>
> [2] Yao, Yifan, et al. "A survey on large language model (llm) security and privacy: The good, the bad, and the ugly." High-Confidence Computing (2024): 100211.

---

> ### Author Response · Authors · 2024-11-24
> **Author Response to Reviewer YKJ1 (2/3)**
>
> ---
> ### **Q2. Why do we use single-time attacks?**
>
> Thank you for the valuable comments. While it is indeed possible to attack TGNNs at multiple timestamps, practical scenarios often impose the limitation of lacking future knowledge about graph evolution. Specifically, **without future knowledge, it is challenging for attackers to determine optimal injection timestamps or how much the attack budget should be used.** As a result, a one-shot attack reflects a more constrained and realistic setup, aligning with the practical challenges faced by adversaries in dynamic graphs.
>
> In our experiments, each attack uses all available knowledge up to the attack timestamp to generate a 'currently optimized' noise. However, as discussed in Section 3 and demonstrated in Section 5, this noise is often nullified quickly due to subsequent changes in the graph.
>
> To evaluate attacks under multiple-time attack setup, we include more results about multiple-time injection in Table R.5 following our setup in Appendix C6, including TGNN performances under TDGIA and our attacks. The attacks are injected right before $t_0, t_5, t_{10}, t_{15}$ with 1% attack budget (i.e., 1% of all nodes) at each time. These results have been added to Section C.6 of the revised paper. As shown, under multiple timestamp attack setups, our attack leads to greater performance degradation in TGNN models against TIGIA.
>
> >**Table R.5. The Accumulated accuracy(accumulated) and batch accuracy(current) across different timestamps in multiple-time TIGIA and MemFreezing Attacks.**
> |        |       |             | $t_1$   | $t_2$   | $t_3$   | $t_4$   | $t_5$   | $t_6$   | $t_7$   | $t_8$   | $t_9$   | $t_{10}$  | $t_{11}$  | $t_{12}$ | $t_{13}$  | $t_{14}$  | $t_{15}$  | $t_{16}$  | $t_{17}$  | $t_{18}$  | $t_{19}$  | $t_{20}$  |
> |--------|-------|-------------|------|------|------|------|------|------|------|------|------|------|------|------|------|------|------|------|------|------|------|------|
> | WiKi   | TDGIA | Current    | 0.86 | 0.88 | 0.92 | 0.87 | 0.92 | 0.91 | 0.94 | 0.95 | 0.89 | 0.93 | 0.87 | 0.95 | 0.82 | 0.87 | 0.87 | 0.94 | 0.89 | 0.95 | 0.90 | 0.82 |
> |        |       | Accumulated  | 0.86 | 0.87 | 0.89 | 0.89 | 0.90 | 0.90 | 0.90 | 0.91 | 0.91 | 0.91 | 0.91 | 0.91 | 0.91 | 0.90 | 0.90 | 0.90 | 0.90 | 0.90 | 0.90 | 0.90 |
> |        | MemFreezing | Current     | 0.90  | 0.87  | 0.90  | 0.92  | 0.86  | 0.89  | 0.88  | 0.93  | 0.96  | 0.89  | 0.76  | 0.85  | 0.82  | 0.85  | 0.79  | 0.88  | 0.87  | 0.82  | 0.87  | 0.88  |
> |        |       | Accumulated  | 0.90  | 0.88  | 0.89  | 0.90  | 0.89  | 0.89  | 0.89  | 0.90  | 0.90  | 0.90  | 0.89  | 0.88  | 0.88  | 0.88  | 0.87  | 0.87  | 0.87  | 0.87  | 0.87  | 0.87  |
> | Reddit | TDGIA | Current     | 0.89 | 0.95 | 0.95 | 0.94 | 0.89 | 0.90 | 0.89 | 0.93 | 0.92 | 0.92 | 0.89 | 0.93 | 0.87 | 0.88 | 0.89 | 0.94 | 0.89 | 0.90 | 0.88 | 0.88 |
> |        |       | Accumulated | 0.89 | 0.91 | 0.92 | 0.92 | 0.92 | 0.92 | 0.91 | 0.91 | 0.91 | 0.91 | 0.91 | 0.91 | 0.91 | 0.91 | 0.91 | 0.91 | 0.91 | 0.91 | 0.90 | 0.90 |
> |        | MemFreezing  | Current     | 0.93  | 0.92  | 0.92  | 0.90  | 0.90  | 0.89  | 0.91  | 0.86  | 0.92  | 0.93  | 0.85  | 0.85  | 0.80  | 0.83  | 0.87  | 0.83  | 0.84  | 0.79  | 0.83  | 0.82  |
> |        |       | Accumulated | 0.93  | 0.92  | 0.92  | 0.92  | 0.91  | 0.91  | 0.91  | 0.90  | 0.90  | 0.91  | 0.90  | 0.90  | 0.89  | 0.89  | 0.89  | 0.88  | 0.88  | 0.88  | 0.87  | 0.87  |
>
> These results show that MemFreezing attack consistently achieves greater performance degradation on TGNN models than baseline methods. In contrast, other attacks, whether one-time or multiple-time, suffer from the effects of future changes and result in limited performance degradation.

---

> ### Author Response · Authors · 2024-11-24
> **Author Response to Reviewer YKJ1 (3/3)**
>
> ---
> ### **Q3. What is the related attack budget on edges?**
>
> Thank you for the question. We would like to clarify that the baseline attacks and MemFreezing are compared using the same attack capabilities. First, to ensure a fair comparison, **all attacks target the same set of victim nodes (with high degrees)**. Second, all benchmarked attacks, including MemFreezing, either inject one-degree nodes or edges into the graph and affect the same number of victim nodes at the time of the attack.
>
> Regarding the second point, MemFreezing specifically targets high-degree nodes by introducing a temporary fake node for each target and creating an event (i.e., an edge) between the fake node and the target. In this way, MemFreezing, like FakeNode, injects nodes with a degree of one into the graph. However, unlike FakeNode, which retains the injected fake nodes and can potentially cause stronger adversarial effects, MemFreezing removes these fake nodes after the attack, minimizing structural changes while inducing long-lasting adversarial effects. Therefore,  given a graph with $V$ nodes and $E$ edges and targeting $N = 0.05V$ victim nodes (i.e., 5% budget), MemFreezing adds $N$ fake edges. Since nodes typically have a degree greater than one, $K = 0.05E > 0.05V = N$, the edge changes are less than 5% edges.
>
> We also clarify this in Appendix C.2 (detailed attack setups) in our revised paper and clarify that our attack injects one event per target node in Section 4.3.
>
> ---
> ### **Q4. Insufficient notations were used.**
>
> We appreciate the reviewer’s feedback regarding the clarity of Sections 3 and 4. To address the concern, we have revised these sections in our updated manuscript to include more consistent notations and formal formulations.

---

> ### Author Response · Authors · 2024-11-25
> **Author Response to Reviewer YKJ1**
>
> Dear Reviewer YKJ1,
>
> Thank you again for your time and valuable feedback. We hope that our responses have addressed your concerns.
>
> We noticed that the confidence score was updated from 3 to 5, but the overall rating remained unchanged. Could we kindly ask if any concerns remain, or if our responses have raised any new questions?
>
> We sincerely look forward to your comments and would be glad to address any additional feedback or questions you may have.
>
> Best,
>
> Authors

---

> > ### Author Response · Authors · 2024-12-02
> > **Author Response to Reviewer YKJ1**
> >
> > Dear Reviewer YKJ1,
> >
> > Thank you again for your valuable feedback and for taking the time to review our work. As the **discussion period is nearing its end**, we wanted to kindly follow up on our earlier message to inquire if there are any remaining concerns or additional feedback you might wish to share. We understand and respect that your time is limited and valuable, and we greatly appreciate the effort you have already dedicated to reviewing our submission.
> >
> > We would be grateful for your input if there are any particular reasons behind the change in the confidence score or additional insights you would like to share. We are committed to addressing any remaining questions or suggestions to further strengthen our work before the discussion period concludes.
> >
> > Thank you once again for your thoughtful engagement, and we look forward to hearing from you.
> >
> > Best regards,
> >
> > Authors

---

> > ### Comment · Reviewer_YKJ1 · 2024-12-02
> > **Response by Reviewer**
> >
> > We thank the authors for their response. However, my major concerns regarding the threat model remain unaddressed. The paper claims to study "practical" attacks but assumes an attacker with white-box knowledge, which is inherently impractical in many real-world scenarios.
> >
> > Regarding the response to Q2 on single-time attacks, the authors state that "without future knowledge, it is challenging for attackers to determine...". This suggests that the attack strategy does not fully leverage the assumed white-box threat model, which further undermines the consistency of the model being considered.
> >
> > Overall, the threat model and the problem formulation remain unclear to me. The use of the term "practical" is misleading, given the unrealistic assumptions made. Additionally, without the stringent white-box threat model, there are several existing published works addressing similar problems but more *practical* (black-box / grey-box knowledge). However, none of them are cited and discussed in the paper.

---

> > > ### Author Response · Authors · 2024-12-03
> > > **Author Response to Reviewer YKJ1**
> > >
> > > Dear Reviewer YKJ1,
> > >
> > > Thank you for your thoughtful feedback and for raising these important concerns.
> > >
> > > - **Definition of White-Box Attacks**
> > >
> > > According to the adversarial machine learning literature, white-box attacks are typically defined as those where the adversary has full knowledge of the model (architecture, parameters, weights, and gradients) [1]. In the context of Graph Neural Networks (GNNs), some works have extended the definition of "white box" to include all information [2]. However, what information beyond model knowledge is unspecified.
> > >
> > > For dynamic graphs and their models, this definition becomes even more complex due to the significance of temporal graph updates. Specifically, there is no consensus on whether or how future graph updates—beyond the time of the attack—should be included in the definition of white-box attacks.
> > >
> > > In our submission, we use the basic white-box setting as in [1]; that is, the white box includes only the model knowledge at the attacking time.
> > >
> > > To avoid the confusion, we plan to clarify this as follows.
> > >
> > > >+ A white-box attack model includes model knowledge and future graph update knowledge.
> > >
> > > >+ A grey-box attack model includes model knowledge only.
> > >
> > > And we adopt the grey-box attack model.
> > >
> > > - **Practicality of Our Attack**
> > >
> > > The discussion on "practicality" in our paper originates from an intriguing research problem: whether an attacker can obtain or predict graph update events after the attack time. This distinction forms the primary difference between our attack model and existing models.
> > >
> > > An attack model incorporating future input knowledge introduces an additional requirement. While this requirement could potentially be met through methods such as AI models, there is no guarantee of success. Furthermore, studies in the literature have shown that the accuracy of predicting future events can be low. For instance, even state-of-the-art TGNN models [3, 4], as advanced spatial-temporal predictors, struggle to accurately forecast the occurrence of edges (i.e., events), let alone retrieve detailed information for attack purposes, such as timestamps, edge features, or related node memories. By removing this requirement from our attack model, we construct a "more practical" attack, though not necessarily "the practical" attack.
> > >
> > > The formal definition of "practicality" remains absent in the literature. Given the evolving nature of the security field, some attacks, once deemed impractical, may later succeed in bypassing system defenses. Therefore, it is more meaningful to assess the relative practicality of different attack models. While we can determine that a known attack is practical, it is impossible to definitively conclude that an unknown attack is impractical.
> > >
> > > - **Request for Clarification on Related Work:**
> > >
> > > Regarding the statement that "there are several existing published works addressing similar problems," could you kindly specify which attacks you are referring to? Based on our review, existing dynamic graph attacks all assume adversaries have full knowledge of the input dynamic graph [5, 6, 7, 8]. We would be glad to discuss any related work that adopts a similar setting on dynamic graphs.
> > >
> > > We sincerely hope this clarifies the focus and contributions of our work. If there are additional concerns or specific references we might have overlooked, we would be grateful for further guidance.
> > >
> > > Thank you again for your valuable feedback and constructive engagement.
> > >
> > > Best regards,
> > >
> > > Authors
> > >
> > >
> > > [1] Chakraborty, Anirban, et al. "Adversarial attacks and defences: A survey." arXiv preprint arXiv:1810.00069 (2018).
> > >
> > > [2] Sun, Lichao, et al. "Adversarial attack and defense on graph data: A survey." IEEE Transactions on Knowledge and Data Engineering 35.8 (2022): 7693-7711.
> > >
> > > [3] Wang, Xuhong, et al. "Apan: Asynchronous propagation attention network for real-time temporal graph embedding." Proceedings of the 2021 international conference on management of data. 2021.
> > >
> > > [4] You, Jiaxuan, Tianyu Du, and Jure Leskovec. "ROLAND: graph learning framework for dynamic graphs." Proceedings of the 28th ACM SIGKDD conference on knowledge discovery and data mining. 2022.
> > >
> > > [5] Chen, Jinyin, et al. "Time-aware gradient attack on dynamic network link prediction." IEEE Transactions on Knowledge and Data Engineering 35.2 (2021): 2091-2102.
> > >
> > > [6] Sharma, Kartik, et al. "Imperceptible adversarial attacks on discrete-time dynamic graph models." NeurIPS 2022 temporal graph learning workshop. 2022.
> > >
> > > [7] Sharma, Kartik, et al. "Temporal dynamics-aware adversarial attacks on discrete-time dynamic graph models." Proceedings of the 29th ACM SIGKDD Conference on Knowledge Discovery and Data Mining. 2023.
> > >
> > > [8] Lee, Dongjin, Juho Lee, and Kijung Shin. "Spear and Shield: Adversarial Attacks and Defense Methods for Model-Based Link Prediction on Continuous-Time Dynamic Graphs." Proceedings of the AAAI Conference on Artificial Intelligence. Vol. 38. No. 12. 2024.

---

### Official Review · Reviewer_xoUn · 2024-11-04

**Soundness:** 4
**Presentation:** 3
**Contribution:** 3
**Rating:** 8
**Confidence:** 4

**Summary:**

This paper explores the challenge of practical adversarial attacks on TGNNs and introduces a novel framework called MemFreezing. The method creates a so-called “frozen” state in node memories by adding fake nodes or edges, which prevents nodes from sensing the graph changes and thus disrupting model predictions. Experimental results show that MemFreezing effectively reduces the performance of TGNNs across various datasets and models, and outperforms existing attack methods.

**Strengths:**

- The paper is written clearly and technically sounded.
- The approach that creates a “frozen” state in node memories by adding fake nodes or edges is interesting.
- The experiments are thorough and comprehensive.

**Weaknesses:**

I have some concerns about the hypothesis of the frozen state as below.

**Questions:**

1. The attacks are assumed to be able to propagate through neighboring nodes and consistently distrupt predictions. However, I concern that changes in graph structure and heterogeneity among nodes and edges might limit the propagation effect. I suggest including specific experiments or analyses to evaluate the robustness of the attack propagation under varying graph dynamics or heterogeneity conditions. For example, it would be helpful to test the attack on graphs with different rates of structural change or varying levels of node/edge heterogeneity. This additional analysis could provide a more comprehensive understanding of the attack’s performance in diverse scenarios.

2. The paper proposes develop surrogate future neighbors using current neighbors, but in practice, it is a bit questionable to use this to reflect future graph changes, especially for those irregular or highly random dynamic graphs. I suggest the authors validate their approach on more irregular or random dynamic graphs. For example, they could test their method on synthetic graphs with varying levels of randomness or analyze how well their surrogate neighbors align with actual future neighbors in their datasets over time.

---

> ### Author Response · Authors · 2024-11-24
> **Author Response to Reviewer xoUn (1/2)**
>
> **We sincerely thank the reviewer for the positive feedback and valuable comments. In response, we clarify the question regarding the propagation of noise in dynamic graphs and provide a quantitative investigation of whether our neighbor simulation scheme remains effective or can be further enhanced under significantly irregular and random cases. We hope our response can help clarify the reviewer's questions.**
>
> ---
> ###  **Q1. Will the propagation of the noises be limited by the changing graph?**
>
> Thank you for the insightful question. We agree that changes in graph structure and heterogeneity among nodes and edges could potentially limit the propagation of attack effects. However, as shown in Figure 8 and Appendix C.13 of our paper, the number of affected victim nodes significantly increases over time, with many nodes becoming frozen. This is achieved through two key mechanisms:
>
> **Targeting High-Degree Nodes:** Noisy events are designed to perturb high-degree nodes at first hand, which act as hubs and influence a large number of neighbors, even as the graph evolves. As shown in Figure 8 and Appendix C.13, targeting high-degree nodes results in nearly twice the number of affected nodes compared to targeting low-degree nodes, making them more effective in real-world scenarios.
>
> **Propagation Through Stable States:** Once a node enters a stable (frozen) state, it continues to affect its future neighbors. This ensures that, even with changes in structure, edge types, or node types, the attack can propagate through alternate routes. The figures and section cited demonstrate that the number of affected nodes grows consistently over time despite the dynamic nature of the graph.
>
> We also add the abovementioned discussion in Section 5.2 in our revised paper.
>
> ---
> ###  **Q2. How does using current neighbors to surrogate future neighbors deal with highly irregular and random graphs? (1/2)**
>
> Thank you for the insightful question. Yes, using the current neighbor cannot ensure that the noise is perfectly solved for considerably irregular and random neighbors. We opt to use current neighbors to surrogate future neighbors since, as observed across most datasets, a node tends to retain highly similar neighbors over time.
>
> To investigate the generalizability of this observation, we further investigate the similarity distribution across diverse datasets (Table R.3) following the same setup as Figure 4(d) in the paper. These results indicate that, **generally, nodes tend to have similar neighbors across diverse datasets**. Hence, using current neighbors provides a reasonable approximation of future graph changes in practice.
>
> > **The distribution of cosine similarities among the ideal frozen states in different nodes in Reddit and Reddit-body datasets.**
> |      Cosine Similarity       | 0.0-0.1 | 0.1-0.2 | 0.2-0.3 | 0.3-0.4 | 0.4-0.5 | 0.5-0.6 | 0.6-0.7 | 0.7-0.8 | 0.8-0.9 | 0.9-1.0 |
> |-------------|---------|---------|---------|---------|---------|---------|---------|---------|---------|---------|
> | Reddit      | 0.0130  | 0.0120  | 0.0120  | 0.0680  | 0.2450  | 0.4170  | 0.1800  | 0.0440  | 0.0060  | 0.0030  |
> | Reddit-Body | 0.0170  | 0.0010  | 0.0140  | 0.0250  | 0.0820  | 0.2630  | 0.4230  | 0.1540  | 0.0820  | 0.0070  |
>
> To investigate if our future neighbor simulation scheme is sufficient to freeze neighbors under irregular or highly random dynamic graphs, we simulate an irregular and random graph on top of the Wikipedia dataset. Specifically, we have victim nodes in the graph connected to nodes with random memories in the future timestamps. We also explored an alternative scheme to investigate whether the heuristic could be further enhanced. Specifically, in this alternative, we simulate nodes' future neighbors using nodes with random memories.

---

> ### Author Response · Authors · 2024-11-24
> **Author Response to Reviewer xoUn (2/2)**
>
> ---
> ###  **Q2. How does using current neighbors to surrogate future neighbors deal with highly irregular and random graphs? (2/2)**
>
> As shown in Table R.4, although resulting in lower similarities, MemFreezing effectively freezes these random neighbors. This demonstrates that our future simulation schemes (i.e., Current Simulation) are effective in irregular setups. The reason behind this is that, in addition to using current neighbors, we also simulate "new future neighbors" with all-zero memories, which further enhance the noise's capability to freeze unseen nodes.
>
> Although the alternative scheme (i.e., Random Simulation) performs better under random neighbor cases (i.e., Noise Future), it shows worse performances in the real cases (i.e., Normal Future). These findings collectively suggest that using current neighbors as surrogates is both practical and effective, even in challenging dynamic graph scenarios.
>
> We also add this discussion in Appendix C.15 in our revised paper and add a pointer to it in Section 4.2.
>
> > **Table R.4. The similarities between victim nodes’ initial noisy memories (at the time of the attack) and themselves’/their subsequent neighbors’ memories in Wikipedia dataset (Normal Future) and its randomized version (Noise Future) under (a) no-attack (i.e., Vanilla), (b) MemFreezing using current neighbor for simulation (i.e., Current Simulation), and MemFreezing using random memory neighbor for simulation (i.e., Random Simulation).**
> |                    |               |       | $t_1$    | $t_2$   | $t_3$   | $t_4$   | $t_5$   | $t_6$   | $t_7$   | $t_8$   | $t_9$   | $t_{10}$  | $t_{11}$  | $t_{12}$  | $t_{13}$  | $t_{14}$  | $t_{15}$  |
> |--------------------|---------------|-------|-------|------|------|------|------|------|------|------|------|------|------|------|------|------|------|
> |     Vanilla |     Noise Future  |     Root  | 0.90 | 0.80 | 0.73 | 0.67 | 0.63 | 0.59 | 0.56 | 0.54 | 0.52 | 0.50 | 0.48 | 0.47 | 0.46 | 0.44 | 0.43 |
> |                  |                   |     1-Hop | 0.17 | 0.15 | 0.14 | 0.13 | 0.12 | 0.12 | 0.11 | 0.11 | 0.11 | 0.10 | 0.10 | 0.10 | 0.10 | 0.10 | 0.10 |
> |                  |                   |     2-Hop | 0.04 | 0.04 | 0.03 | 0.03 | 0.03 | 0.03 | 0.03 | 0.03 | 0.03 | 0.02 | 0.03 | 0.03 | 0.03 | 0.03 | 0.03 |
> |                  |     Normal Future |     Root  | 0.90 | 0.84 | 0.81 | 0.78 | 0.76 | 0.74 | 0.72 | 0.72 | 0.71 | 0.69 | 0.67 | 0.68 | 0.67 | 0.67 | 0.66 |
> |                  |                   |     1-Hop | 0.22 | 0.26 | 0.28 | 0.28 | 0.28 | 0.27 | 0.26 | 0.25 | 0.24 | 0.23 | 0.22 | 0.21 | 0.20 | 0.19 | 0.18 |
> |                  |                   |     2-Hop | 0.07 | 0.07 | 0.06 | 0.06 | 0.06 | 0.06 | 0.06 | 0.06 | 0.05 | 0.05 | 0.05 | 0.06 | 0.06 | 0.06 | 0.06 |
> | Current Simulation| Noise Future  | Root  | 0.96  | 0.94 | 0.91 | 0.89 | 0.89 | 0.87 | 0.86 | 0.85 | 0.84 | 0.83 | 0.82 | 0.82 | 0.81 | 0.81 | 0.80 |
> |                    |               | 1-Hop | 0.24  | 0.38 | 0.47 | 0.54 | 0.58 | 0.62 | 0.64 | 0.67 | 0.69 | 0.70 | 0.72 | 0.73 | 0.74 | 0.75 | 0.75 |
> |                    |               | 2-Hop | -0.03 | 0.10 | 0.22 | 0.31 | 0.37 | 0.41 | 0.47 | 0.49 | 0.50 | 0.54 | 0.57 | 0.58 | 0.60 | 0.62 | 0.64 |
> |                    | Normal Future | Root  | 0.99  | 0.97 | 0.97 | 0.96 | 0.95 | 0.94 | 0.93 | 0.93 | 0.93 | 0.93 | 0.93 | 0.93 | 0.93 | 0.92 | 0.92 |
> |                    |               | 1-Hop | 0.51  | 0.57 | 0.64 | 0.67 | 0.71 | 0.75 | 0.78 | 0.80 | 0.82 | 0.84 | 0.86 | 0.87 | 0.88 | 0.89 | 0.90 |
> |                    |               | 2-Hop | 0.06  | 0.21 | 0.34 | 0.44 | 0.51 | 0.58 | 0.64 | 0.67 | 0.71 | 0.74 | 0.77 | 0.79 | 0.81 | 0.82 | 0.84 |
> | Random Simulation    | Noise Future  | Root  | 0.98  | 0.96 | 0.95 | 0.93 | 0.92 | 0.91 | 0.91 | 0.90 | 0.91 | 0.90 | 0.89 | 0.89 | 0.88 | 0.89 | 0.88 |
> |                    |               | 1-Hop | 0.33  | 0.45 | 0.51 | 0.56 | 0.57 | 0.61 | 0.65 | 0.68 | 0.71 | 0.73 | 0.74 | 0.76 | 0.78 | 0.79 | 0.81 |
> |                    |               | 2-Hop | -0.05 | 0.08 | 0.21 | 0.31 | 0.39 | 0.45 | 0.51 | 0.55 | 0.59 | 0.62 | 0.65 | 0.68 | 0.70 | 0.73 | 0.75 |
> |                    | Normal Future | Root  | 0.98  | 0.97 | 0.95 | 0.93 | 0.93 | 0.93 | 0.92 | 0.91 | 0.88 | 0.89 | 0.89 | 0.88 | 0.87 | 0.88 | 0.88 |
> |                    |               | 1-Hop | 0.38  | 0.50 | 0.54 | 0.53 | 0.60 | 0.65 | 0.69 | 0.72 | 0.74 | 0.76 | 0.78 | 0.80 | 0.81 | 0.82 | 0.83 |
> |                    |               | 2-Hop | -0.03 | 0.14 | 0.28 | 0.38 | 0.45 | 0.52 | 0.58 | 0.62 | 0.65 | 0.67 | 0.70 | 0.72 | 0.74 | 0.76 | 0.77 |

---

> > ### Comment · Reviewer_xoUn · 2024-11-24
> >
> > Thanks for the response. One minor suggestion is that, for Q2, the authors should highlight this assumption to address potential concerns from the audience. The reviewer has no further comments and will remain the score.

---

> > > ### Author Response · Authors · 2024-11-24
> > > **Author Response to Reviewer xoUn**
> > >
> > > We would like to thank you again for your time and your positive rating. This is a great affirmation of our work. We highlight the assumption in our response and will further revise our paper to make it clearer to the audience.

---

### Official Review · Reviewer_7jhv · 2024-11-04

**Soundness:** 2
**Presentation:** 2
**Contribution:** 3
**Rating:** 5
**Confidence:** 4

**Summary:**

The paper proposes an adversarial attack, termed MemFreezing, for Temporal Graph Neural Networks (TGNNs). MemFreezing selects pairs of victim nodes and crafts accompanying messages to update the memory of TGNNs s.t. the memory resides in an unrecoverable and update-resistant state. The authors empirically demonstrate that the heuristics underpinning MemFreezing are effective and, as a result, MemFreezing has a long-lasting impact on the attacked TGNN.

**Strengths:**

1. The paper is well-written, and I found it easy to follow the general gist.
1. The authors propose the first attack on TGNN without knowledge of the future data.
1. The authors empirically verify that the TGNN remains effected by the attack for a considerably long time.

**Weaknesses:**

1. The message of the paper strongly depends on the perspective of "practical attacks," without the authors specifying what properties "practicality" entails and why  "practical attacks" are a relevant research topic. Is it the goal of the authors to provide a ready-to-use adversarial attack for real-world adversaries? I hope not. I know that (for reasons unknown to me) it is often advocated that "practicality" is important for adversarial attacks on graph-structured data. While this stance is not necessarily attributed to the work at hand, it still should be discussed prominently (i.e., introduction). Also, the truthfulness of statements like "Attackers can only acquire knowledge up to the attack’s timestamp." depends on the perspective. It is not uncommon to assume an adversary with oracle-like capabilities to study the worst-case performance w.r.t. small meaningless perturbations.
1. Ideally, the authors quantify how the attack compares to an "impractical" attack with perfect knowledge. Although this seems out of scope for a rebuttal. I do not expect experiments up on this.
1. The authors could be more explicit that their attack leverages heuristics like "cross-freeing". Also, statements like "In this state, nodes can no longer sense graph changes or carry information [...]" are overclaiming without the authors providing proof that this was the case.
1. There are many other choices of attack capabilities that are not well discussed and are arguably "impractical" as well. For example, (a) the attacker has perfect knowledge about the model and all data up to $t_0$. (b) MemFreezing chooses the highest-degree nodes and then their highest-degree neighbors. In most graphs, it is very unlikely that a real-world adversary would have access to such node pairs. (c) limiting the message values by the min/max of the features (i.e., [-1, +1]) seems not very realistic/practical either. (d) focusing on inserting all adversarial messages at a single point in time (right before test) is arbitrary and likely to be detected by trivial anomaly detection methods. Results in C6 are not very convincing since FakeNodes appears to be the weakest attack.

I am willing to increase my score if the weaknesses are addressed.

Minor:
1. Using "sample" for a topk procedure was confusing to me (Section 4.3)
1. The procedure of obtaining the "nodes' ideal frozen state" was not clear from the main text.
1. It would be better to apply \text etc. to, e.g., subscripts like "L_{mse}"

**Questions:**

1. Why do the benchmarked attacks compare in their attack capabilities? For example, as far as I understand, inserting new nodes will result in very low-degree nodes, while MemFreezing will attack the nodes with the highest degree.
1. How does MemFreezing compare to TGDIA applied multiple points in time? (C6)
1. How is "Cross-freezing" achieved if attacking at multiple points in time?

---

> ### Author Response · Authors · 2024-11-24
> **Author Response to Reviewer 7jhv (1/4)**
>
> **We sincerely appreciate the valuable comments and insights from the reviewer. In response, we carefully respond to the reviewer’s questions and revise the paper accordingly, including a more detailed discussion and clarification on the definition and value of the practical attack setup, a clearer explanation of the attacker’s capabilities, additional results and discussions on multiple-time attacks, and adjustments to certain overclaiming statements. We hope our response and revisions can help alleviate the reviewer's concern.**
>
> ---
>
> ### **Q1. What is the practicality of the proposed MemFreezing Attack, and why is it valuable to explore TGNN attacks under a more practical setting?**
>
> Thank you very much for the thoughtful comment! We agree that it is crucial to clarify the value of studying adversarial attacks under practical constraints and to define what 'practicality' entails in our paper.
>
> Regarding 'practicality', our MemFreezing attack is not intended as a ready-to-use attack for real-world adversaries, which would require additional capabilities, such as crawling online data (e.g., input graph and victim model) and forging fake users or behaviors (e.g., injecting adversarial noise). Instead, compared to prior works that assume oracle-like capabilities, our attack adopts a more realistic setup by operating under limited knowledge. This limited-knowledge setup makes MemFreezing relatively more practical and closer to scenarios that could occur in real-world applications.
>
> While studying worst-case scenarios with oracle-like knowledge is valuable for understanding the upper bounds of vulnerability, attacks under real-world constraints can **reveal distinct flaws in TGNNs that might remain hidden in idealized settings**. For instance, while all-knowing attacks can demonstrate the most harmful perturbations, they do not necessarily expose how fragile the memory mechanism is under more feasible constraints. By exploring attacks with limited knowledge, our work aims to uncover potential threats that are more relevant to real-world deployments and encourage the community to address these practical challenges.
>
> To address the confusion and better highlight the relevance of practicality, we also add the above discussion in Section 1 to make these points clearer.
>
> ---
>
> ### **Q2. More clarification on Cross-Freezing.**
>
> Thank you for the valuable comments. We agree that the frozen nodes (i.e., nodes affected by the attack) are not completely static or unchanging. Rather, these nodes exhibit significantly higher memory similarity before and after updates at subsequent timestamps, which compromises their responsiveness to surrounding changes. We have revised the related statements in the paper to better reflect this observation.
>
> Our conclusion is supported by Figure 7 in the original submission. As shown in Figure 7 (left), under the MemFreezing attack, the memories of frozen nodes maintain high similarity, averaging over 0.92 cosine similarity. In contrast, without the attack, memory similarities among unfrozen nodes decrease significantly over time, with less than 0.20 on average. This demonstrates that MemFreezing induces nodes to remain stable (highly similar) over future updates, limiting their ability to adapt to changes in the graph.
>
> Additionally, we clarify that our attack leverages heuristics like 'cross-freezing' in Section 1 in our revised paper to explicitly mention this feature.

---

> ### Author Response · Authors · 2024-11-24
> **Author Response to Reviewer 7jhv (2/4)**
>
> ---
> ### **Q3. Choices of Attacker’s Capabilities.**
>
> Thank you for your detailed and thoughtful comments. We greatly appreciate your insights, as they help us clarify and improve our work. Below, we respond to each of the points in detail and explain how they are considered in our revised paper.
>
> ---
> - **Is white-box attack and up-to-attack knowledge practical?**
>
> It is practical to assume knowledge of the model and all graph data up to $t_0$​ in many dynamic graph applications, as these graph information is often publicly accessible. For instance, platforms like Wikipedia, Reddit, Meta, or X maintain dynamic graphs that can be crawled from official or related tracing websites, enabling adversaries to reconstruct input graph data with reasonable accuracy.
>
> In terms of model parameters, many TGNN architectures and pre-trained models are open-sourced, making them readily available to adversaries. Furthermore, methods such as insider threats or model extracting [1][2] can be employed to extract model parameters when the model itself is not publicly available. These factors collectively make the white-box attack setup relatively more practical and realistic in many real-world scenarios.
>
> However, future knowledge (e.g., data or updates occurring after $t_0$​) is inherently harder to access due to its temporal and evolving nature. Current methods offer no feasible way to reliably predict future changes in graph structure or labels. As such, we focus on the more challenging constraint of using only past knowledge for attacks while adhering to the white-box setup for model parameters.
>
> We clarify this point further in Section 3.1 of the revised paper to address this concern explicitly.
>
>
> [1] Oliynyk, Daryna, Rudolf Mayer, and Andreas Rauber. "I know what you trained last summer: A survey on stealing machine learning models and defences." ACM Computing Surveys 55.14s (2023): 1-41.
>
> [2] Yao, Yifan, et al. "A survey on large language model (llm) security and privacy: The good, the bad, and the ugly." High-Confidence Computing (2024): 100211.
>
> ---
> - **How do we access to the highest-degree nodes?**
>
> We do not directly access the highest-degree nodes; instead, for each high degree node, we induce their memory into noisy states by injecting an noisy event from a noisy node to it. The noisy node can be removed after the attack.
>
> For instance, in Reddit, we could conduct the attack in the following steps: (1) First, we recognize those most-viewed/most-commented Reddit posts (i.e., highest-degree nodes); (2) Second, we create a new user node (with noisy memory features) and use it to make a comment on those Reddit posts (i.e., injecting a noisy event ). And the user can be removed (e.g., unregister) after attack.
>
>
> ---
> - **Why do we limit the noisy message range between -1/+1?**
>
> As detailed in Appendix C.2., we limit the message range between -1/+1  since -1 and 1 are the theoretical minimum and maximum values of the clean messages. The messages in TGNNs are usually memories of the nodes updated from previous timestamps. The memory updater in these TGNNs are usually GRUCells or RNNCells, which have tanh activation functions right before the outputs. Therefore, all features of these messages (i.e., memories) should be within the range of -1 and 1 as the minimum and maximum values of the activation functions (i.e., tanh). Hence, using -1 and 1 produces noisy memories and, consequently, noisy messages similar to those of the clean messages in the graph.

---

> ### Author Response · Authors · 2024-11-24
> **Author Response to Reviewer 7jhv (3/4)**
>
> ---
> - Multiple timestamp attack
>
> We include more results about multiple-time injection in Table R.1 following our setup in Appendix C6, including TGNN performances under TDGIA and our attacks. The attacks are injected right before $t_0, t_5, t_{10}, t_{15}$ with 1% attack budget (i.e., 1% of all nodes) at each time. As shown, under multiple timestamp attack setups, our attack leads to greater performance degradation in TGNN models against TIGIA. We also add these results to Section C.6 of the revised paper.
>
> >**Table R.1. The Accumulated accuracy(accumulated) and batch accuracy(current) across different timestamps in multiple-time TIGIA and MemFreezing Attacks.**
> |        |       |             | $t_1$   | $t_2$   | $t_3$   | $t_4$   | $t_5$   | $t_6$   | $t_7$   | $t_8$   | $t_9$   | $t_{10}$  | $t_{11}$  | $t_{12}$ | $t_{13}$  | $t_{14}$  | $t_{15}$  | $t_{16}$  | $t_{17}$  | $t_{18}$  | $t_{19}$  | $t_{20}$  |
> |--------|-------|-------------|------|------|------|------|------|------|------|------|------|------|------|------|------|------|------|------|------|------|------|------|
> | WiKi   | TDGIA | Current    | 0.86 | 0.88 | 0.92 | 0.87 | 0.92 | 0.91 | 0.94 | 0.95 | 0.89 | 0.93 | 0.87 | 0.95 | 0.82 | 0.87 | 0.87 | 0.94 | 0.89 | 0.95 | 0.90 | 0.82 |
> |        |       | Accumulated  | 0.86 | 0.87 | 0.89 | 0.89 | 0.90 | 0.90 | 0.90 | 0.91 | 0.91 | 0.91 | 0.91 | 0.91 | 0.91 | 0.90 | 0.90 | 0.90 | 0.90 | 0.90 | 0.90 | 0.90 |
> |        | MemFreezing | Current     | 0.90  | 0.87  | 0.90  | 0.92  | 0.86  | 0.89  | 0.88  | 0.93  | 0.96  | 0.89  | 0.76  | 0.85  | 0.82  | 0.85  | 0.79  | 0.88  | 0.87  | 0.82  | 0.87  | 0.88  |
> |        |       | Accumulated  | 0.90  | 0.88  | 0.89  | 0.90  | 0.89  | 0.89  | 0.89  | 0.90  | 0.90  | 0.90  | 0.89  | 0.88  | 0.88  | 0.88  | 0.87  | 0.87  | 0.87  | 0.87  | 0.87  | 0.87  |
> | Reddit | TDGIA | Current     | 0.89 | 0.95 | 0.95 | 0.94 | 0.89 | 0.90 | 0.89 | 0.93 | 0.92 | 0.92 | 0.89 | 0.93 | 0.87 | 0.88 | 0.89 | 0.94 | 0.89 | 0.90 | 0.88 | 0.88 |
> |        |       | Accumulated | 0.89 | 0.91 | 0.92 | 0.92 | 0.92 | 0.92 | 0.91 | 0.91 | 0.91 | 0.91 | 0.91 | 0.91 | 0.91 | 0.91 | 0.91 | 0.91 | 0.91 | 0.91 | 0.90 | 0.90 |
> |        | MemFreezing  | Current     | 0.93  | 0.92  | 0.92  | 0.90  | 0.90  | 0.89  | 0.91  | 0.86  | 0.92  | 0.93  | 0.85  | 0.85  | 0.80  | 0.83  | 0.87  | 0.83  | 0.84  | 0.79  | 0.83  | 0.82  |
> |        |       | Accumulated | 0.93  | 0.92  | 0.92  | 0.92  | 0.91  | 0.91  | 0.91  | 0.90  | 0.90  | 0.91  | 0.90  | 0.90  | 0.89  | 0.89  | 0.89  | 0.88  | 0.88  | 0.88  | 0.87  | 0.87  |
>
>
> It is also worth mentioning that we prioritize one-shot attacks because, **without knowing future knowledge, it is challenging to determine when to inject attacks and how much of the attack budget should be used.**
>
>
> ---
> ### **Q4. Do benchmarked attacks result in low-degree nodes while MemFreezing results in highest-degree nodes?**
>
> Thank you for the question. We would like to clarify that the baseline attacks and MemFreezing are compared using the same attack capabilities. First, to ensure a fair comparison, **all attacks target the same set of victim nodes**. Second, all benchmarked attacks, including MemFreezing, inject either one-degree nodes or edges into the graph and affect the same number of victim nodes at the time of the attack.
>
> Regarding the second point, MemFreezing specifically targets high-degree nodes by introducing a temporary fake node for each target and creating an event (i.e., an edge) between the fake node and the target. In this way, **MemFreezing, like FakeNode, injects nodes with a degree of one into the graph**. However, unlike FakeNode, which retains the injected fake nodes and can potentially cause stronger adversarial effects, MemFreezing removes these fake nodes after the attack, minimizing structural changes while inducing long-lasting adversarial effects.
>
> Thus, while MemFreezing targets high-degree nodes, it leverages low-degree nodes through the temporary introduction of fake nodes, offering an effective yet lightweight attack strategy. In contrast, all baseline attacks target the same high-degree nodes while introducing more changes.
>
> We also clarify this in Appendix C.2 (detailed attack setups) in our revised paper and clarify that our attack injects one event per target node in Section 4.3.

---

> ### Author Response · Authors · 2024-11-24
> **Author Response to Reviewer 7jhv (4/4)**
>
> ---
> ### **Q5. How is "Cross-freezing" achieved if attacking at multiple points in time?**
>
> Thank you for the question. To achieve 'cross-freezing' when attacking at multiple points in time, we ensure that a victim node and its two supporting neighbors (i.e., three connected victim nodes) are attacked in close temporal proximity. Specifically, for each group of victim nodes (i.e., mutually connected three nodes), the MemFreezing either attacks them in a single injection or within consecutive attack rounds. By minimizing the time between attacks on these nodes, their memories are less likely to diverge before mutual support is established, enabling them to reinforce each other and remain in stable states after the attack.
>
> To evaluate the effectiveness of cross-freezing under multiple-time attack cases, we investigate the similarities between victim nodes' initial noisy memories (at the time of the attack) and themselves'/their subsequent neighbors' memories in MemFreezing under one-time attack setup and multiple-times attack setup (following the setup in Figure 7 in our paper).
>
> As shown in Table R.2, even when attacks occur at multiple points in time, the victim nodes still exhibit high similarity in their memory states during subsequent updates as the one-time attack, demonstrating that the cross-freezing mechanism is still effective under multiple-attack cases. We have added these results to Appendix C.6 in our revised paper to clarify this mechanism.
>
> > **Table R.2. The similarities between victim nodes' initial noisy memories (at the time of the attack) and themselves'/their subsequent neighbors' memories in MemFreezing under one-time attack setup and multiple-times attack setup.**
> |        |            | t1   | t2   | t3   | t4   | t5   | t6   | t7   | t8   | t9   | t10  | t11  | t12  | t13  | t14  | t15  |
> |--------|------------|------|------|------|------|------|------|------|------|------|------|------|------|------|------|------|
> | Wiki   | Multi-time | 0.98 | 0.95 | 0.94 | 0.93 | 0.97 | 0.94 | 0.93 | 0.93 | 0.92 | 0.96 | 0.95 | 0.93 | 0.91 | 0.91 | 0.90 |
> |        | One-Time   | 0.99 | 0.97 | 0.97 | 0.96 | 0.95 | 0.94 | 0.93 | 0.93 | 0.93 | 0.93 | 0.93 | 0.93 | 0.93 | 0.92 | 0.92 |
> | Reddit | Multi-time | 0.95 | 0.94 | 0.93 | 0.92 | 0.94 | 0.93 | 0.92 | 0.91 | 0.91 | 0.93 | 0.93 | 0.92 | 0.91 | 0.89 | 0.89 |
> |        | One-Time   | 0.98 | 0.96 | 0.94 | 0.94 | 0.94 | 0.94 | 0.93 | 0.92 | 0.91 | 0.91 | 0.92 | 0.91 | 0.90 | 0.91 | 0.90 |
>
>
> ---
> ###  **Q6. Comments on the presentation.**
>
> Thanks for the valuable suggestion. We made the following changes in our revised paper:
>
> >- We change “sample” in Section 4.3 to “select”.
> >- We revise Section 4.1 to explain how we get nodes’ ideal frozen state more clearly.
> >- We apply \text on loss subscripts as suggested, in terms of node indices, we keep their math format to distinguish them from the other texts.

---

> > ### Author Response · Authors · 2024-11-27
> > **Author Response to Reviewer 7jhv**
> >
> > Dear Reviewer 7jhv,
> >
> > Thanks for your time and reviewing efforts! We appreciate your constructive comments.
> >
> > We provide suggested results in the authors' response, including:
> >
> > - Clarify the value of studying adversarial attacks under a practical setup.
> >
> > - Justify the choices based on the attacker's capability.
> >
> > - Discuss the resulting node degrees after different attacks and clarify that all attacks target the same victim node-set.
> >
> > - Provide comparison results with TIGIA on multiple-time attacks.
> >
> > - Analyze the cross-freezing performances under multiple-time attack setup.
> >
> > We hope our responses have answered your questions. It would be our great pleasure if you would consider updating your review or score. We would be glad to address any additional feedback or questions you may have.
> >
> > Best,
> >
> > Authors

---

> > > ### Comment · Reviewer_7jhv · 2024-11-28
> > >
> > > I thank the authors for their careful and detailed response.
> > >
> > > I think I still do not fully understand what the authors refer to as "practicality" and why this was the most important angle for research on TGNN's robustness. While I agree that the attack strategies can differ between full knowledge and limited knowledge, attack capabilities under limited knowledge are a subset of those at full knowledge. In the real world, there will anyways always be an arms race that is very specific to the circumstances of model deployment, etc. And it feels to me as if the authors picked some arbitrary constraints to make the attack "more practical."
> > >
> > > Rest assured, I do not want to nitpick for reasons against this work, as I think it has its merit. However, I find the accompanying discussion and justification still somewhat artificial/superficial, although the revision has already improved in that regard. At least one of the other reviewers also seems to share this opinion. I have raised the score accordingly.

---

> > > > ### Author Response · Authors · 2024-11-28
> > > > **Author Response to Reviewer 7jhv**
> > > >
> > > > Thank you very much for your thoughtful feedback and for raising the score. We sincerely appreciate your recognition of the merit in our work and your acknowledgment of the improvements in the revised discussion.
> > > >
> > > > We understand your concern regarding the definition and justification of "practicality". To conduct ideal adversarial attacks on dynamic graphs, an adversary typically requires knowledge of three aspects at the time of the attack: (a) the TGNN model details, (b) all past events in the dynamic graph, and (c) future events in the graph. Our study specifically focuses on attack scenarios where the adversary lacks knowledge of (c) future events.
> > > >
> > > > Rather than arbitrarily selecting constraints, we believe that studying adversarial attacks under limited knowledge of future events deserves focused attention because real-world adversaries frequently operate under this constraint. Unlike acquiring model details (via insider threats or model extraction) or existing graph information (via web crawling), accessing every future change in an evolving dynamic graph is particularly challenging. This key distinction motivated us to focus on this specific subset of constraints (i.e., without future knowledge), which we believe is significantly more probable in real-world attacks.
> > > >
> > > > While "practicality" can vary depending on the deployment context, we argue that examining attacks under these realistic constraints reveals vulnerabilities in TGNNs that are often overlooked in idealized, full-knowledge settings. By addressing this overlooked area, our work aims to contribute to a more comprehensive understanding of TGNN robustness.
> > > >
> > > > We also agree that further exploration of other practical constraints is valuable, particularly in black-box settings where attackers lack access to model parameters or internal states. Incorporating these scenarios is definitely a valuable direction for future research, and your insights have helped shape our plans for further exploration.
> > > >
> > > > Once again, thank you for your detailed evaluation and constructive feedback. Your insights have been invaluable in strengthening our manuscript, and we are grateful for your thoughtful engagement.

---

### Comment · Area_Chair_yA2B · 2024-11-28

I would like to encourage the reviewers to engage with the author's replies if they have not already done so. At the very least, please
acknowledge that you have read the rebuttal.

---

### Meta-Review · Area_Chair_yA2B · 2024-12-21

**Metareview:**

The paper proposes an adversarial attack on temporal GNNs. This is apparently the first attack that assumes no knowledge of future data. The method creates a so-called “frozen” state in node memories via perturbations, which prevents memory updates and consequently reduces performance.

The issue of "practicality" was raised and discussed with the reviewers. While indeed not assuming future knowledge does make the attack more "practical" the authors still assume perfect knowledge about the model and all data up to $t_0$. Therefore, it is not clear how to interpret the results from the attack because it test neither worst-case performance nor real-world practical performance. Other choices (e.g. highest-degree nodes) were also questioned. The authors reply addressing these concerns was not fully convincing, and I tend to agree with the assessment of Reviewers 7jhv and YKJ1.

In the future, I suggest that the authors to rethink the motivation for the attack and provide a stronger justification for the threat model, potentially de-emphasising the importance of whether the attack is practical or not. Going beyond the white-box setting (e.g. with surrogate models) and comparing with attacks that do have future knowledge to quantify how much this knowledge is important would be good steps towards improving the paper.

**Additional Comments On Reviewer Discussion:**

Two reviewers questioned the "practicality" of the attacks and the motivation behind the threat model. The authors did not include additional experiments beyond the white-box setting even though it was raised as one of the concerns. Given how easy it is to train a surrogate model I think there is no excuse to not include such variants, especially given the focus on practicality. One of the reviewer raised the score to 5, but the other Reviewer kept the score at 3 since their major concerns regarding the threat model remained unaddressed.

---

### Decision · Program_Chairs · 2025-01-22

Reject